# You Only Communicate Once:
# One-shot Federated Low-Rank Adaptation of MLLM

**Binqian Xu**[1]**, Haiyang Mei**[2]**, Zechen Bai**[2]**, Jinjin Gong**[1]**, Rui Yan**[1]**, Guo-Sen Xie**[1]**, Yazhou Yao**[1]**,**
**Basura Fernando**[3]**, and Xiangbo Shu**[1,*]
[1]Nanjing University of Science and Technology    [2]Show Lab, National University of Singapore
[3]Institute of High-Performance Computing, A*STAR
https://github.com/1xbq1/FedMLLM

## Abstract

Multimodal Large Language Models (MLLMs) with Federated Learning (FL) can
quickly adapt to privacy-sensitive tasks, but face significant challenges such as
high communication costs and increased attack risks, due to their reliance on multi-
round communication. To address this, One-shot FL (OFL) has emerged, aiming
to complete adaptation in a single client-server communication. However, existing
adaptive ensemble OFL methods still need more than one round of communication,
because correcting heterogeneity-induced local bias relies on aggregated global
supervision, meaning they still do not achieve true one-shot communication. In
this work, we make the first attempt to achieve true one-shot communication for
MLLMs under OFL, by investigating whether implicit (*i.e.*, initial rather than
aggregated) global supervision alone can effectively correct local training bias. Our
key finding from the empirical study is that imposing directional supervision on
local training substantially mitigates client conflicts and local bias. Building on this
insight, we propose YOCO, in which directional supervision with sign-regularized
LoRA B enforces global consistency, while sparsely regularized LoRA A preserves
client-specific adaptability. Experiments demonstrate that YOCO cuts communi-
cation to ∼0.03% of multi-round FL while surpassing those methods in several
multimodal scenarios and consistently outperforming all one-shot competitors.

## 1   Introduction

Multimodal Large Language Models (MLLMs) have rapidly advanced in recent years [29], driven by
their powerful capabilities and broad applicability, achieving impressive performance across diverse
multimodal tasks through fine-tuning [57, 1, 61, 62]. However, in privacy-sensitive scenarios such
as healthcare, centralized data collection for fine-tuning is often impractical [43, 37, 52], prompting
growing interest in adapting MLLMs via Federated Learning (FL) [54, 53, 63]. Although Parameter-
Efficient Fine-Tuning (PEFT) methods [19, 23] help reduce communication costs in these approaches,
federated fine-tuning still involves transmitting large amounts of parameters across multiple rounds,
resulting in high system overhead and increased security risks [44, 51], as shown in Figure 1 (a).

One-shot FL (OFL), involving a single round of communication, is an advantageous FL paradigm,
with the communication cost equivalent to that of a single model parameter [32, 42]. Specifically,
without relying on additional data or generative models, adaptive ensemble OFL methods [42, 3]
achieve good performance by effectively correcting local training biases using supervision derived
from aggregated global weights. "Local training bias" refers to overfitting local data during client-side
training, leading to spurious correlations caused by data heterogeneity and degrading generalization.

---

*Corresponding author

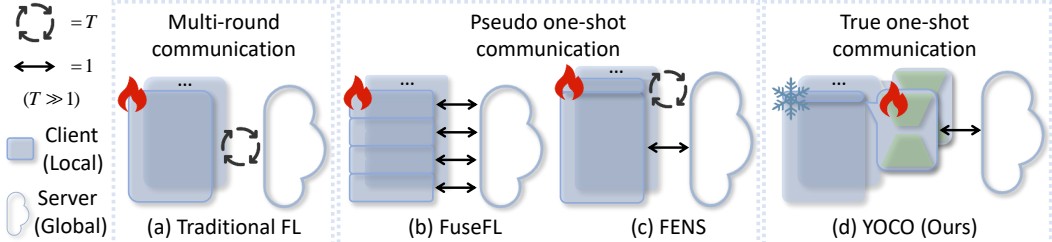

Figure 1: (a) Traditional FL involves multi-round communication, resulting in high system overhead and increased security risks. (b) OFL: FuseFL [42] is a pseudo one-shot communication method that runs beyond a single round, guided by global supervision from bottom-up aggregated blocks. (c) OFL: FENS [3] is a pseudo one-shot communication method that operates for more than one round, guided by global supervision from multi-round lightweight aggregators. (d) YOCO is a true one-shot communication method, guided by implicit global supervision. Without additional data or generative models, FuseFL and FENS are the two most representative adaptive ensemble OFL methods.

To mitigate this, "global supervision", represented by aggregated weights containing shared knowledge, is used to guide local training. However, the generation and propagation of global supervision contradict the single-round (one-shot) communication requirement, as shown in Figure 1 (b) and (c).

In contrast, as shown in Figure 1 (d), during one-shot federated low-rank adaptation of MLLMs, all client models share the same pre-trained weights. Unlike the aggregated "global supervision" mentioned above, pre-trained weights, defined as "implicit global supervision", contain more universal and generalized knowledge. With true one-shot communication to reduce overhead and attack risks, using implicit global supervision to guide local LoRA training results in the core problem shifting to,

*Can implicit global supervision in true one-shot comm. correct local bias?*

To answer this question, we explore bias correction for local LoRA training, supervised by the magnitude and direction of implicit global supervision. Here, each LoRA [23] weight consists of an A matrix (LoRA A) and a B matrix (LoRA B). For weights, the magnitude refers to the absolute value, while the direction refers to the sign of each parameter. Based on our empirical study, we draw the following two important conclusions: Direction constraints cause less interference with optimization than magnitude constraints. Constraining only the direction of LoRA B, combined with prior initialization, further reduces conflicts and helps correct local LoRA training bias (Section 3).

Based on the above conclusions, we propose YOCO (Figure 2), a true one-shot communication method. In YOCO, the initial direction across clients follows the prior direction (Figure 2 (a)), using matrix decomposition of pre-trained weights as initial weights of LoRA (for both LoRA A and B). During local training (Figure 2 (b)), progressive regularization guides the updates of LoRA B toward the direction of implicit supervision, by measuring the disparity in the signs of parameters, thus mitigating aggregation conflicts from model discrepancies. While this consistency constraint benefits aggregation, it may hinder adaptation to client-specific knowledge. In response to this, we introduce sparse regularization to LoRA A, encouraging the learning of critical client-specific knowledge, followed by noise-free aggregation weighted by principal components (Figure 2 (c)). Compared to multi-round methods, YOCO uses only ∼0.03% of the communication cost, even outperforming them in both aligned and missing modal scenarios on the CrisisMMD dataset. In the true one-shot communication setting, YOCO consistently outperforms all baselines, proving its effectiveness.

Our main contributions can be summarized as follows,

- **True one-shot communication**. We are the first to explore true one-shot communication for MLLMs with OFL, addressing the key challenge of whether implicit global supervision can effectively correct local training bias in this true one-shot communication setting.

- **Balance consistency and adaptability**. We propose YOCO, which uses implicit global supervision to correct local training bias. Derived from this supervision, prior initialization and sign-based regularization on LoRA B ensure global consistency, while sparsity regularization and principal component-weighted aggregation on LoRA A enhance local adaptability.

- **Communication-aware Comparison**. Compared to multi-round communication methods, YOCO requires only about 0.03% of the communication cost and even outperforms these methods in some

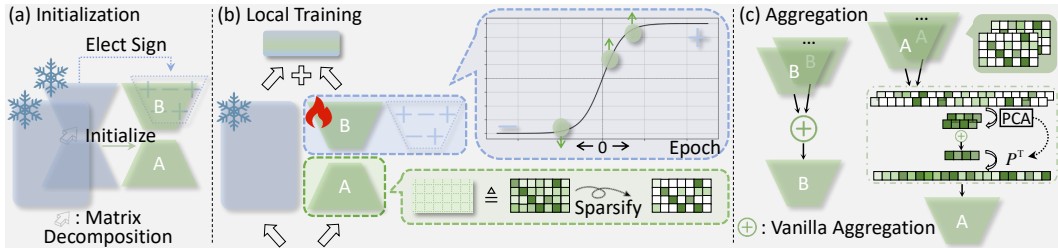

Figure 2: Illustration of YOCO. (a) Initialization: LoRA weights are initialized from the matrix decomposition of pre-trained weights, providing a shared global prior direction and the sign (i.e., $+/-$) of each parameter for LoRA B. (b) Local Training: In addition to the original task loss, LoRA B is incrementally regularized by the prior sign to maintain global consistency across clients, while LoRA A is sparsely regularized with the $\ell_1$ norm to strengthen client-specific knowledge and minimize irrelevant noise. (c) Aggregation: LoRA B uses vanilla averaging due to lower cross-client conflict, while LoRA A adopts principal component-weighted aggregation to enhance adaptability.

multimodal scenarios. When compared to multiple true one-shot communication baselines, YOCO delivers the best performance across all multimodal scenarios.

## 2 Background

One-shot Federated Learning (OFL) involves a single communication round, with a cost equivalent to the size of one model's parameter [32, 42]. Compared to traditional FL, OFL offers notable advantages in privacy and efficiency [16]. Existing OFL methods are generally categorized into four types: static ensemble, adaptive ensemble, knowledge distillation, and generative models. Knowledge distillation and generative models [58, 10, 21] are closely linked. Specifically, knowledge distillation includes both data [66, 40] and model distillation [13, 64, 15], with data sourced from both public and generative model-generated synthetic data. Unlike the methods above, static and adaptive ensemble methods offer greater flexibility in combining with others. Early approaches focused on static ensemble [14, 24, 45], analyzing local clients' model statistics to derive global parameters. Recently, adaptive ensemble methods [42, 3] leverage aggregated global supervision to correct local bias, and improve generalization. In contrast to pseudo one-shot communication methods such as FuseFL [42] and FENS [3], YOCO first enables true one-shot communication for federated low-rank adaptation of MLLMs via implicit global supervision. More detailed related works are provided in the appendix.

## 3 Empirical Motivation

**Magnitude vs. Direction.** To investigate the key question "*Can implicit global supervision in true one-shot communication correct local training bias?*", we first decouple implicit global supervision into two distinct constraint components—magnitude and direction—and apply them separately to guide local LoRA training. As illustrated in Figure 3, we randomly select a client and present the training loss curves of LoRA under three different settings: (i) the original training without any constraints; (ii) training with only a magnitude constraint, where the $L2$ norm is used to measure the distance between LoRA weights and the implicit global supervision; and (iii) training with only a direction constraint, where the per-element sign differences are minimized to align the direction

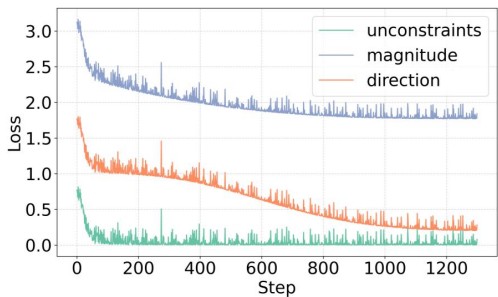

Figure 3: Loss curves under three LoRA training setups: unconstraints, magnitude constrain, and direction constrain. Direction constrain shows lower loss than magnitude constrain and gradually approaches that of the unconstrained setup.

of LoRA weights with that of the implicit global supervision. In the design of the constraint, the implicit global supervision is projected onto the dimensions of LoRA through matrix decomposition. The experimental results demonstrate that applying the direction constraint yields a faster loss descent

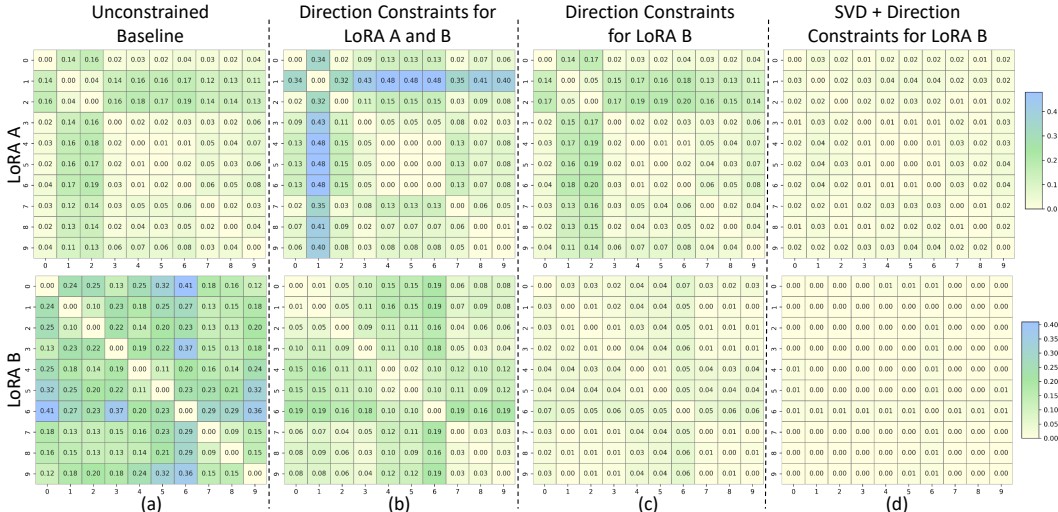

Figure 4: Conflict matrices across clients: (a) Unconstrained baseline, (b) Direction constraints for LoRA A and B, (c) Direction constraints for LoRA B, and (d) SVD + Direction constraints for LoRA B. The x- and y-axes denote client indices; lighter colors indicate lower conflict and bias. In summary, SVD initialization, combined with direction constraints only on LoRA B, most effectively reduces conflicts and corrects local training biases under the same constraint level.

and eventually converges to a level close to the unconstrained case, outperforming the setting with only a magnitude constraint. This indicates that the direction constraint can effectively convey the implicit global supervision without significantly disturbing the original LoRA training process.

**Insights on Direction Constraints for LoRA Fine-Tuning.** LoRA updates are restricted to a low-rank subspace, limiting their ability to capture the full parameter space of full-parameter fine-tuning. Aligning these updates with the pre-trained weight direction allows LoRA to follow the adaptation path of full fine-tuning while remaining lightweight. Crucially, it is the direction, not the magnitude, that provides effective implicit global supervision to guide local updates. Prior studies support this: LoRA-GA [46] uses SVD-based initialization to preserve pre-trained directions and align initial gradients; LoRA-Pro [48] enforces this alignment throughout training; and FR-LoRA [56] applies a similar approach in federated settings, yielding consistent gains. Collectively, these results show that maintaining alignment with pre-trained directions is key to narrowing the gap between LoRA and full-parameter fine-tuning, highlighting the central role of directional constraints in LoRA fine-tuning.

**LoRA A or LoRA B.** To assess the effectiveness of direction constraints in correcting the training bias of LoRA (including both LoRA A and LoRA B) across different clients, we introduce the conflict matrix as a visualization tool. The conflict matrix consists of pairwise conflict rates $\rho$, which measure the proportion of parameters with opposing signs, reflecting weight disagreements between clients [55]. When $k_i$ client signs are consistent on parameter $i$, the average pairwise conflict rate is $\bar{\rho} = \frac{2}{MN(N-1)} \sum_i k_i(N - k_i)$, and the overall conflict rate is $C = \frac{1}{MN} \sum_i \min(k_i, N - k_i)$. Here, $M = \text{card}(\Delta \mathbf{W})$ represents the number of parameters in $\Delta \mathbf{W}$ and $N$ is the total number of clients. $C$ is positively correlated with $\bar{\rho}$, i.e., $C \propto \bar{\rho}$, and serves as an upper bound approximation of $\bar{\rho}$,

$$\bar{\rho} = \frac{2N}{N-1} \left[ C - \frac{1}{MN^2} \sum_i [\min(k_i, N - k_i)]^2 \right] \tag{1}$$

Assume each parameter's magnitude follows a distribution with mean $\mu$ and variance $\sigma^2$. The expected value and variance of the global LoRA weights $\Delta \mathbf{W_g}$ are as follow,

$$\mathbb{E}[\Delta \mathbf{W_g}] = (1 - 2C)\mu; \quad \text{Var}(\Delta \mathbf{W_g}) = \frac{1}{N}[\sigma^2 + 4C(1 - C)\mu^2] \tag{2}$$

As $C \in [0, 0.5]$ decreases, $|1 - 2C|$ increases, amplifying the expected magnitude of the global weights and clarifying their direction. Meanwhile, $4C(1 - C)\mu^2$ decreases, reducing variance and stabilizing the update, effectively correcting local training bias across clients. For example, in a tug-of-war, $C$ is the proportion of force opposing the target. As $C$ decreases, more people align their

force with the target, stabilizing the process and correcting individual force biases. The detailed derivation is in the appendix. Lemma 3.1 concludes the proof, as follows,

**Lemma 3.1** (Less Conflict, Less Bias). *A decrease in $\bar{\rho}$, along with a reduction in $C$, an increase in the expected magnitude $|\mathbb{E}[\Delta \mathbf{W_g}]|$, and a decrease in the variance $\mathrm{Var}(\Delta \mathbf{W_g})$, helps stabilize the global update direction, thereby effectively mitigating local training bias.*

For LoRA A and B, we initially attempt to impose direction constraints on both simultaneously. However, as shown in Figure 4 (b), constraining both LoRA A and B introduces excessive restrictions, which in turn increases the direction conflicts of LoRA A across clients, as it limits A's projection space and forces clients into a narrow subspace, making A more sensitive to local data variations. Therefore, we choose to constrain only one of LoRA A or LoRA B. Since the conflict between client LoRA B is greater in the unconstrained baseline (Figure 4 (a)), we decide to constrain only LoRA B. As shown in Figure 4 (c), the conflict among client LoRA A remains almost unchanged, while the conflict among LoRA B is significantly reduced. Finally, inspired by [47, 49], we initialize LoRA A and B using the Singular Value Decomposition (SVD) of the implicit global supervision to provide a prior direction for all clients. Under the guidance of direction constraints, conflicts are further mitigated (Figure 4 (d)). Notably, excessive bias correction may be counterproductive, potentially impairing LoRA's ability to learn client-specific knowledge. Thus, balancing bias correction with local exploration is essential.

# 4 YOCO

Based on the above motivation, we propose YOCO for true one-shot communication in federated low-rank adaptation of MLLMs. Specifically, YOCO consists of initialization, local training with sign-based consistency and noise-free sparse regularization, followed by aggregation. The direction supervision based on implicit global supervision enables the SVD initialization of LoRA and the sign-based consistency regularization of LoRA B. To balance bias correction with local exploration, noise-free sparse regularization of LoRA A is introduced, followed by principal component-weighed aggregation. The illustration of YOCO is shown in Figure 2.

**Notation and Definitions.** In the classical FL framework, such as FedAvg [34], the system typically consists of $N$ clients, represented by the set $\mathcal{N} = \{n\}_{n=1}^{N}$. Each client possesses a local private dataset $\mathcal{D}_n$ and conducts local training on a model parameterized by $\boldsymbol{\theta}$, optimized using a task-specific loss function $\mathcal{L}_s$ [54]. After completing local training in the $t^{th}$ round ($t \in [1, T], T \gg 1$), the system randomly selects a subset of clients $\mathcal{N}_t \subset \mathcal{N}$ to upload their models $\{\boldsymbol{\theta}_i^t, i \in \mathcal{N}_t\}$ to the server for aggregation, resulting in a global model, i.e., $\boldsymbol{\theta}^{t+1} := \sum_{i \in \mathcal{N}_t} p_i \boldsymbol{\theta}_i^t$, where $p_i = \frac{|\mathcal{D}_i|}{\sum_{j \in \mathcal{N}_t} |\mathcal{D}_j|}$. Unlike FL, OFL [32] limits client-server communication to a single round ($T = 1$), reducing privacy risks and communication costs. In one-shot federated low-rank adaptation for MLLMs, most parameters are frozen and correspond to the pre-trained weights $\mathbf{W}$, while only the small sets of LoRA parameters are trained and uploaded for aggregation, i.e., the updated weights $\Delta \mathbf{W} = \mathbf{B}\mathbf{A}$. Here, $\mathbf{B}$ and $\mathbf{A}$ represent the weights of LoRA B and LoRA A, respectively.

**Initialization.** Following [49], we use SVD of the pre-trained weights to initialize LoRA A and B,

$$\mathbf{U}, \boldsymbol{\Sigma}, \mathbf{V}^{\top} = \mathtt{SVD}(\mathbf{W})$$
$$\mathbf{U}_r = \mathbf{U}_{[:,1:r]}, \boldsymbol{\Sigma}_r = \boldsymbol{\Sigma}_{[1:r]}, \mathbf{V}_r^{\top} = \mathbf{V}_{[1:r,:]}^{\top} \quad (3)$$
$$\mathbf{A} = \mathbf{A}^0 = \mathtt{diag}(\sqrt{\boldsymbol{\Sigma}_r})\mathbf{V}_r^{\top}, \mathbf{B} = \mathbf{B}^0 = \mathbf{U}_r\mathtt{diag}(\sqrt{\boldsymbol{\Sigma}_r})$$

where $r$ denotes the rank of LoRA, and $\mathtt{diag}(\cdot)$ represents the operation of constructing a diagonal matrix. $\mathbf{A}^0$ and $\mathbf{B}^0$ are the initialized weights of LoRA A and B, respectively. It is worth noting that while using SVD to decompose pre-trained weights for LoRA initialization is common [65, 49], in this true one-shot task, the focus of this prior initialization is to provide different clients with a more informative and consistent starting point and direction for local LoRA training.

**Sign-based Consistency Regularization.** In the initialization phase, $\mathbf{B}^s$ is obtained by extracting the sign of $\mathbf{B}^0$, i.e., $\mathbf{B}^s = \mathrm{sign}(\mathbf{B}^0)$. Inspired by [55], during aggregation, the proportion of elements with reversed signs in the parameters reflects the conflict between models. Smaller conflicts facilitate less bias (**Lemma 3.1**). Therefore, to correct the local training bias for improved one-shot aggregation, we present a direction-alignment regularization method that measures the distance between the signs

of $\mathbf{B}$ and $\mathbf{B}^s$, called Sign-based Consistency Regularization. Since $\mathrm{sign}(\mathbf{B})$ is non-differentiable, we use the smooth approximation $\tanh(\cdot)$ instead. The regularization $\mathcal{R}_{\mathrm{sign}}$ is computed as follows,

$$\mathcal{R}_{\mathrm{sign}} = \frac{\|\tanh(\gamma\mathbf{B}) - \mathbf{B}^s\|_2}{\|\mathbf{B}^s\|_2 + \varepsilon} \tag{4}$$

where $\gamma$ controls the smoothness of the soft sign function, and $\varepsilon$ is a small positive number ($1e - 6$) used to prevent division by zero. Notably, only the direction of $\mathbf{B}$ is aligned with $\mathbf{W}$ here. Simultaneously constraining the directions of both $\mathbf{A}$ and $\mathbf{B}$ would impose excessive restrictions, thereby increasing the direction conflicts of $\mathbf{A}$ across different clients, as shown in Figure 4.

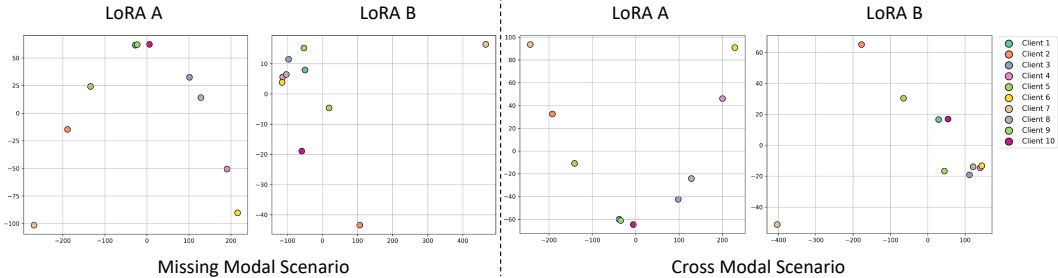

Figure 5: PCA-based Visualization. After local training with loss $\mathcal{L}$, we use PCA to visualize LoRA A and B across clients. LoRA A is more dispersed, while LoRA B is more clustered, suggesting that LoRA B captures shared knowledge, whereas LoRA A reflects client-specific information.

**Noise-free Sparse Regularization.** Aligning the global direction of $\mathbf{B}$ reduces conflicts between clients and aids aggregation. However, the specific information in clients' private datasets often differs from the general features learned during pretraining. To balance global consistency and local adaptability, we enhance client-specific knowledge through LoRA A, using the simple and effective $\ell_1$ norm for noise-free sparse regularization $\mathcal{R}_{\mathrm{sparse}} = \lambda \times \|\mathbf{A}\|_1$. Where $\lambda$ controls the degree of sparsity enhancement. The total loss of the local training is $\mathcal{L} = \mathcal{L}_s + \mathcal{R}_{\mathrm{sign}} + \mathcal{R}_{\mathrm{sparse}}$. After local training with loss $\mathcal{L}$, the regularization terms encourage LoRA B to learn shared knowledge, while LoRA A captures client-specific features. To illustrate this, we visualize the distribution of LoRA A and B across different clients using Principal Component Analysis (PCA), as shown in Figure 5. The visualization reveals that the more dispersed LoRA A across clients reflects stronger client-specific information, while the more clustered LoRA B suggests a tendency toward shared knowledge.

**Principal Component-weighted Aggregation.** As shown in Figure 5, LoRA B, which captures shared information, is more concentrated and suited for vanilla aggregation, while LoRA A, capturing client-specific information, is more dispersed and should maintain its diversity during aggregation. Inspired by the PCA visualization, we present a principal component-weighted aggregation method for LoRA A across clients. The process is as follows,

$$\mathbf{Z} = \mathrm{PCA}(\mathbf{A}'), \mathbf{Z} \in \mathbb{R}^{N \times k}; \quad \mathbf{A}' = [\boldsymbol{a}_1, \boldsymbol{a}_2, \ldots, \boldsymbol{a}_N] \in \mathbb{R}^{N \times d}, \boldsymbol{a} = \mathrm{vec}(\mathbf{A}) \in \mathbb{R}^d$$
$$\mathbf{A}_g = \mathbf{Z}_g \cdot \mathbf{P}^\top, \mathbf{P}^\top \in \mathbb{R}^{k \times d}; \quad \mathbf{Z}_g = \sum_{i=1}^{N} \frac{|\mathcal{D}_i|}{\sum_{j=1}^{N} |\mathcal{D}_j|} \mathbf{Z}_i, \mathbf{Z}_g \in \mathbb{R}^k \tag{5}$$

where $\mathbf{A}'$ is obtained by flattening and stacking the LoRA A matrices from all clients, and $\mathbf{P}^\top$ denotes the transpose of the principal-component matrix. $\mathbf{Z}_i$ represents the top-$k$ PCA coordinates of client $i$'s LoRA A, and $\mathbf{P}$ is the PCA loading matrix (written as $\mathbf{P}^\top$ in the paper) used to map the weighted average $\mathbf{Z}_g$ back to the original parameter space. $\mathbf{A}_g$ denotes global weights from principal component-weighted aggregation. $\mathbf{B}_g$ is obtained via vanilla aggregation. The aggregation phase occurs on the server, where $\mathbf{A}_g$ and $\mathbf{B}_g$ merge with the pre-trained weights for evaluation.

## 5 Experiments

### 5.1 Experimental Setup

**Datasets.** To evaluate YOCO's effectiveness in true one-shot communication for MLLM adaptation, we use five public multimodal datasets: Hateful-Memes [25], CrisisMMD [2], VQA-RAD [27],

Table 1: Comparison of YOCO with multi-round methods and one-shot baselines across aligned ($\alpha$) and missing ($\beta$) modal scenarios on Hateful-Memes and CrsisMMD. YOCO with $1\times1\times1$ incurs only $\sim$0.03% of communication cost of multi-round methods ($25$(rounds)$\times28$(layers)$\times4$(LoRAs) or $40\times28\times4$), yet outperforms them in all CrisisMMD scenarios, and surpasses all one-shot baselines.

| Method | Comm. cost | Hateful-Memes | | | | | | Comm. cost | CrisisMMD | | | | | |
|---|---|---|---|---|---|---|---|---|---|---|---|---|---|---|
| | | $\alpha$=5.0 | $\alpha$=1.0 | $\alpha$=0.5 | $\beta$=30% | $\beta$=40% | $\beta$=50% | | $\alpha$=5.0 | $\alpha$=1.0 | $\alpha$=0.5 | $\beta$=30% | $\beta$=40% | $\beta$=50% |
| FedYogi | $25\times28\times4$ | 75.74 | 72.50 | 74.32 | 76.58 | 76.75 | 75.12 | $40\times28\times4$ | 60.95 | 60.82 | 61.46 | 60.07 | 56.76 | 53.78 |
| FedAdam | $25\times28\times4$ | 74.02 | 73.24 | 72.97 | 76.82 | 77.28 | 75.58 | $40\times28\times4$ | 61.80 | 59.12 | 60.56 | 59.54 | 57.64 | 54.82 |
| FedAvgM | $25\times28\times4$ | 74.08 | 72.94 | 64.22 | 60.62 | 57.90 | 57.34 | $40\times28\times4$ | 58.28 | 56.87 | 54.22 | 32.64 | 32.97 | 31.72 |
| FedAdagrad | $25\times28\times4$ | 74.53 | 73.34 | 71.80 | 71.97 | 70.64 | 70.59 | $40\times28\times4$ | 60.77 | 60.43 | 60.21 | 56.96 | 57.48 | 55.55 |
| Local | $0\times1\times1$ | 68.84 | 66.75 | 66.69 | 66.66 | 66.24 | 66.31 | $0\times1\times1$ | 54.62 | 47.05 | 37.27 | 36.96 | 36.36 | 36.34 |
| FedAdam | $1\times1\times1$ | 68.85 | 67.31 | 68.15 | 67.45 | 67.32 | 67.33 | $1\times1\times1$ | 51.27 | 46.15 | 44.65 | 42.13 | 42.81 | 39.57 |
| FedAvgM | $1\times1\times1$ | 69.06 | 69.64 | 68.27 | 69.20 | 68.58 | 67.57 | $1\times1\times1$ | 61.52 | 59.72 | 55.17 | 50.64 | 42.90 | 47.05 |
| FedAdagrad | $1\times1\times1$ | 69.24 | 68.16 | 68.17 | 67.64 | 67.35 | 67.35 | $1\times1\times1$ | 33.90 | 33.82 | 33.42 | 32.50 | 32.23 | 30.31 |
| FedAvg | $1\times1\times1$ | 64.72 | 70.44 | 68.19 | 69.21 | 68.58 | 66.97 | $1\times1\times1$ | 61.57 | 59.01 | 46.06 | 50.57 | 41.75 | 46.96 |
| Mean | $1\times1\times1$ | 69.16 | 68.15 | 67.28 | 67.49 | 66.69 | 66.09 | $1\times1\times1$ | 62.47 | 60.50 | 60.05 | 59.70 | 58.37 | 59.55 |
| Ensemble | $1\times1\times1$ | 68.42 | 67.89 | 68.99 | 68.22 | 68.21 | 68.03 | $1\times1\times1$ | 62.13 | 59.88 | 53.71 | 56.43 | 56.74 | 54.46 |
| Combine | $1\times1\times1$ | 69.06 | 69.98 | 68.79 | 69.31 | 68.79 | 67.58 | $1\times1\times1$ | 62.38 | 61.90 | 60.14 | 58.88 | 59.23 | 57.66 |
| YOCO$_{Init}$ | $1\times1\times1$ | 68.90 | 70.18 | 69.60 | 68.81 | 69.03 | 69.03 | $1\times1\times1$ | 63.23 | 63.05 | 60.98 | 58.77 | 59.90 | 57.63 |
| YOCO$_{Init\&B}$ | $1\times1\times1$ | 69.61 | 70.42 | 70.42 | 68.89 | 69.21 | 69.33 | $1\times1\times1$ | 63.58 | 63.17 | 61.12 | 59.90 | 59.94 | 57.77 |
| YOCO$_{Init\&BA}$ | $1\times1\times1$ | 69.92 | 69.20 | 70.64 | 69.05 | 69.76 | 69.45 | $1\times1\times1$ | 63.51 | 62.27 | 61.73 | 60.12 | 60.06 | 57.57 |
| YOCO$_{Init\&B\bar{A}}$ | $1\times1\times1$ | 70.14 | 70.76 | 70.65 | 69.48 | 69.39 | 69.96 | $1\times1\times1$ | 63.69 | 63.66 | 61.28 | 59.95 | 59.71 | 58.10 |

SLAKE [30], and MedAlpaca [18] (the last three are medical datasets). Following [54], we conduct experiments under aligned, missing, cross, and hybrid modal scenarios with non-IID partitioning from a Dirichlet distribution ($\alpha = \{5.0, 1.0, 0.5\}$), considering multimodal heterogeneity ($\beta = \{30\%, 40\%, 50\%\}$, I-T=\{3-7, 5-5, 7-3\}, and $p = \{80\%, 70\%, 60\%\}$). Evaluation metrics are AUC for Hateful-Memes, F1 for CrisisMMD, and GPT-4 [1]-assessed accuracy for the medical datasets.

**Baselines.** To ensure a fair comparison in the true one-shot communication setting, we construct a series of "$1\times1\times1$" baselines, representing minimal communication cost with a single round, layer, and LoRA. "FedYogi", "FedAdam", and "FedAdagrad" [38] use dynamic optimizers, while "FedAvgM" [22] applies momentum. Unlike "Mean", "FedAvg" [34] incorporates sample-size weighting. "Ensemble" combines LoRA votes from all clients. "Combine" denotes the different aggregations for LoRA A and B. Local in "$0\times1\times1$" represents the average result from independent client training. We experiment with four YOCO variants: YOCO$_{init}$ (initialization only), YOCO$_{init\&B}$ (initialization with sign-based consistency regularization), YOCO$_{init\&BA}$ (noise-free sparse regularization built on YOCO$_{init\&B}$), and YOCO$_{init\&B\bar{A}}$ (adding principal component-weighted aggregation to YOCO$_{init\&BA}$).

**Configurations.** Similar to [54], we adopt MiniCPM-V-2_6-int4 [62] as the default version of MLLMs. Only the last layer's q_proj is fine-tuned using LoRA [23], with a rank of 8, a lora_alpha of 8, and a dropout rate of 0.05. Under the default setting, the total number of clients is 10, with each client trained for 10 epochs. We report the best performance of different baselines at both the $5^{th}$ and $10^{th}$ epochs. For the Hateful-Memes dataset, the initial learning rate is set to 2e-5, while for the other datasets, it is set to 2e-4. A cosine learning rate scheduler is used, with a warmup ratio of 1%. The per-device training batch size is set to 1, and the gradient accumulation steps are set to 4. As shown in Table 4, we also conduct experiments with different MLLM versions and varying total numbers of clients. Details of $\gamma$ and $\lambda$ are in the appendix. All experiments are performed on NVIDIA A40.

## 5.2 Experimental Results

**Main results.** As shown in Tables 1 to 3, YOCO is comprehensively evaluated on four public datasets. We can see that: 1) YOCO outperforms all one-shot baselines and consumes only $\sim$0.03% of the communication cost of multi-round methods, even surpassing them in some CrisisMMD and medical scenarios; 2) YOCO$_{init\&B}$ outperforms YOCO$_{Init}$ in most cases, validating the effect of regularization, while the superior performance of YOCO$_{init\&BA}$ and YOCO$_{init\&B\bar{A}}$ highlights the importance of enhancing local adaptability; 3) YOCO with $1\times1\times1$ also lowers computation cost.

**Loss curves for YOCO variants.** As shown in Figure 6, we present the training loss curves of multiple YOCO variants on three randomly selected clients. Compared to YOCO$_{Init}$, YOCO$_{init\&B}$, which incorporates sign-based consistency regularization for LoRA B, converges to a similar loss

Table 2: Comparison of YOCO with multi-round methods and one-shot baselines in cross (I-T) and hybrid ($p$) modal scenarios on Hateful-Memes and CrsisMMD. YOCO with $1\times1\times1$ uses only ∼0.03% of communication cost of multi-round methods ($25$(rounds)$\times28$(layers)$\times4$(LoRAs) or $40\times28\times4$), yet performs comparably in some CrisisMMD scenarios and exceeds all one-shot baselines.

| Method | Comm. cost | Hateful-Memes I:3-T:7 | I:5-T:5 | I:7-T:3 | p=80% | p=70% | p=60% | Comm. cost | CrisisMMD I:3-T:7 | I:5-T:5 | I:7-T:3 | p=80% | p=70% | p=60% |
|---|---|---|---|---|---|---|---|---|---|---|---|---|---|---|
| FedYogi | 25×28×4 | 68.71 | 73.93 | 75.94 | 68.41 | 68.55 | 71.52 | 40×28×4 | 49.30 | 55.53 | 59.20 | 59.73 | 63.04 | 58.96 |
| FedAdam | 25×28×4 | 65.71 | 74.44 | 74.95 | 72.17 | 70.12 | 72.36 | 40×28×4 | 53.17 | 56.62 | 59.22 | 60.02 | 61.01 | 59.93 |
| FedAvgM | 25×28×4 | 58.20 | 61.57 | 57.40 | 61.55 | 60.01 | 58.93 | 40×28×4 | 32.61 | 38.62 | 33.46 | 46.26 | 42.77 | 39.44 |
| FedAdagrad | 25×28×4 | 71.26 | 72.48 | 71.57 | 68.77 | 72.96 | 69.26 | 40×28×4 | 48.02 | 49.97 | 51.13 | 57.58 | 58.91 | 53.84 |
| Local | 0×1×1 | 66.04 | 66.93 | 67.23 | 65.61 | 66.42 | 66.44 | 0×1×1 | 32.76 | 33.88 | 35.51 | 37.41 | 36.04 | 37.78 |
| FedAdam | 1×1×1 | 66.75 | 67.24 | 67.85 | 66.82 | 67.85 | 67.65 | 1×1×1 | 32.41 | 33.16 | 36.12 | 41.94 | 38.36 | 37.12 |
| FedAvgM | 1×1×1 | 66.68 | 68.89 | 69.28 | 67.63 | 67.16 | 67.35 | 1×1×1 | 45.21 | 50.38 | 51.67 | 54.92 | 46.26 | 47.54 |
| FedAdagrad | 1×1×1 | 66.85 | 66.75 | 66.66 | 67.74 | 67.96 | 67.16 | 1×1×1 | 28.27 | 29.94 | 29.49 | 32.72 | 32.05 | 31.08 |
| FedAvg | 1×1×1 | 66.48 | 68.49 | 68.69 | 67.43 | 67.96 | 67.35 | 1×1×1 | 45.01 | 51.20 | 51.51 | 55.13 | 45.53 | 47.38 |
| Mean | 1×1×1 | 66.22 | 68.03 | 68.02 | 67.17 | 67.63 | 68.44 | 1×1×1 | 44.07 | 47.89 | 52.15 | 56.61 | 54.31 | 55.48 |
| Ensemble | 1×1×1 | 67.94 | 68.25 | 68.26 | 67.93 | 68.45 | 68.34 | 1×1×1 | 51.48 | 52.84 | 55.69 | 56.93 | 56.31 | 57.60 |
| Combine | 1×1×1 | 66.48 | 69.83 | 69.10 | 67.13 | 67.56 | 68.16 | 1×1×1 | 42.55 | 47.19 | 52.24 | 56.24 | 54.11 | 55.03 |
| YOCO$_{Init}$ | 1×1×1 | 67.76 | 68.72 | 69.73 | 68.14 | 67.86 | 68.85 | 1×1×1 | 52.22 | 53.50 | 55.60 | 58.13 | 55.75 | 58.58 |
| YOCO$_{Init\&B}$ | 1×1×1 | 68.04 | 69.44 | 69.63 | 68.23 | 68.07 | 68.94 | 1×1×1 | 51.59 | 53.78 | 56.12 | 57.96 | 57.98 | 58.61 |
| YOCO$_{Init\&BA}$ | 1×1×1 | 68.14 | 69.83 | 68.74 | 69.13 | 68.36 | 68.84 | 1×1×1 | 52.40 | 56.55 | 55.01 | 58.77 | 59.06 | 58.26 |
| YOCO$_{Init\&B\bar{A}}$ | 1×1×1 | 68.85 | 69.85 | 70.16 | 67.95 | 68.58 | 69.09 | 1×1×1 | 43.33 | 47.51 | 50.33 | 58.43 | 54.99 | 54.76 |

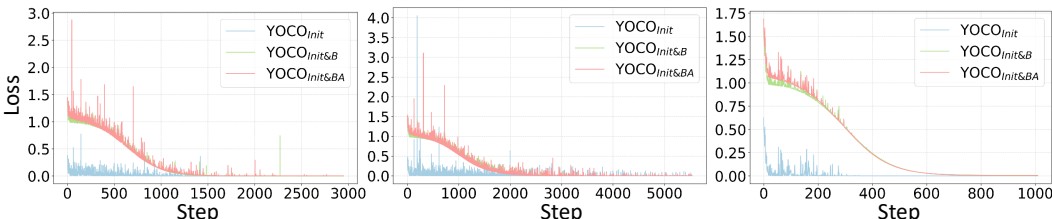

Figure 6: Training loss curves of YOCO$_{Init}$, YOCO$_{init\&B}$, and YOCO$_{init\&BA}$ (YOCO$_{init\&B\bar{A}}$) from three randomly selected clients. Sign-based regularization on LoRA B results in a final loss similar to that without regularization, while sparse regularization on LoRA A has little impact on the training loss.

Table 3: Comparison of YOCO with multi-round methods and one-shot baselines in cross (I-T) and hybrid ($p$) modal scenarios on the medical datasets. YOCO with the $1\times1\times1$ setting incurs only ∼0.02% of the communication cost of multi-round methods ($50$(rounds)$\times28$(layers)$\times4$(LoRAs)), yet outperforms them in most scenarios, and consistently surpasses all one-shot baselines in "Overall".

| Method | Comm. cost | Scen. | VQA-RAD Open | Closed | Overall | SLAKE Open | Closed | Overall | Scen. | VQA-RAD Open | Closed | Overall | SLAKE Open | Closed | Overall |
|---|---|---|---|---|---|---|---|---|---|---|---|---|---|---|---|
| Local | 0×28×4 | I:3-T:7 | 42.74 | 69.97 | 59.16 | 58.71 | 65.83 | 61.63 | I:5-T:5 | 43.86 | 70.04 | 59.64 | 58.19 | 65.19 | 60.98 |
| FedYogi | 50×28×4 | | 45.25 | 68.02 | 58.98 | 53.70 | 66.15 | 58.67 | | 44.42 | 71.14 | 60.53 | 52.55 | 63.04 | 56.74 |
| FedYogi | 1×1×1 | | 43.58 | 67.65 | 58.09 | 62.32 | 61.36 | 61.94 | | 42.46 | 71.69 | 60.09 | 60.41 | 61.72 | 60.94 |
| FedAvgM | 1×1×1 | | 42.46 | 68.01 | 57.87 | 61.92 | 60.41 | 61.32 | | 42.46 | 66.18 | 56.76 | 59.54 | 63.04 | 60.94 |
| YOCO$_{Init\&B\bar{A}}$ | 1×1×1 | | 46.37 | 69.12 | 60.09 | 63.51 | 60.41 | 62.27 | | 44.13 | 73.16 | 61.64 | 60.89 | 66.27 | 63.04 |
| Local | 0×28×4 | I:7-T:3 | 41.90 | 71.69 | 59.87 | 59.22 | 62.32 | 60.46 | p=80% | 41.06 | 71.87 | 59.64 | 52.23 | 67.34 | 58.26 |
| FedYogi | 50×28×4 | | 41.90 | 73.53 | 60.98 | 52.78 | 60.41 | 55.83 | | 44.13 | 70.96 | 60.31 | 53.74 | 64.00 | 57.83 |
| FedYogi | 1×1×1 | | 45.25 | 69.85 | 60.09 | 61.05 | 64.47 | 62.42 | | 40.22 | 71.69 | 59.20 | 59.14 | 64.11 | 61.13 |
| FedAvgM | 1×1×1 | | 44.13 | 70.96 | 60.31 | 59.86 | 62.68 | 60.98 | | 42.46 | 72.43 | 60.53 | 57.39 | 62.56 | 59.46 |
| YOCO$_{Init\&B\bar{A}}$ | 1×1×1 | | 48.60 | 71.69 | 62.53 | 61.69 | 66.63 | 63.66 | | 46.93 | 69.85 | 60.75 | 58.82 | 64.71 | 61.17 |
| Local | 0×28×4 | p=70% | 45.25 | 69.85 | 60.09 | 53.02 | 65.67 | 58.07 | p=60% | 46.93 | 70.96 | 61.08 | 56.36 | 63.52 | 59.22 |
| FedYogi | 50×28×4 | | 43.02 | 72.06 | 60.54 | 54.46 | 64.60 | 58.50 | | 46.64 | 73.90 | 63.08 | 52.43 | 64.00 | 57.05 |
| FedYogi | 1×1×1 | | 46.37 | 68.38 | 59.65 | 60.33 | 62.68 | 61.27 | | 43.02 | 70.59 | 59.65 | 58.35 | 65.19 | 61.08 |
| FedAvgM | 1×1×1 | | 45.81 | 68.38 | 59.42 | 56.60 | 64.59 | 59.79 | | 46.37 | 73.16 | 62.53 | 57.00 | 63.64 | 59.65 |
| YOCO$_{Init\&B\bar{A}}$ | 1×1×1 | | 45.25 | 72.79 | 61.86 | 60.25 | 63.64 | 61.60 | | 45.81 | 73.90 | 62.75 | 60.02 | 65.67 | 62.27 |

level. Introducing noise-free sparse regularization on LoRA A in YOCO$_{init\&BA}$ (or YOCO$_{init\&B\bar{A}}$) on top of YOCO$_{init\&B}$ has minimal impact on the loss convergence trend.

Table 4: Comparison of different lightweight MLLMs under different numbers of total clients (N = 10, N = 20, N = 30). YOCO with the 1×1×1 setting incurs only ∼0.03% of the communication cost of multi-round methods (25(rounds)×28(layers)×4(LoRAs)), yet outperforms them in N = 30 of V-2_6 MLLM and N = 20 of V-2_5 MLLM, consistently surpassing all one-shot baselines.

| Method | Comm. cost | MLLMs | Total Clients | | | MLLMs | Total Clients | | |
|---|---|---|---|---|---|---|---|---|---|
| | | | N = 10 | N = 20 | N = 30 | | N = 10 | N = 20 | N = 30 |
| Local | 0×28×4 | | 66.80 | 64.46 | 63.50 | | 67.02 | 63.21 | 63.26 |
| FedAdam | 25×28×4 | | 75.58 | 69.70 | 66.91 | | 72.44 | 68.82 | 71.16 |
| Local | 0×1×1 | | 66.31 | 65.05 | 65.47 | | 67.12 | 62.71 | 65.20 |
| FedAdam | 1×1×1 | MiniCPM-V-2_6$_{int4}$ | 67.33 | 68.05 | 67.29 | MiniCPM-Llama3-V-2_5$_{int4}$ | 68.62 | 67.80 | 69.60 |
| FedAvgM | 1×1×1 | | 67.57 | 67.69 | 65.97 | | 68.51 | 67.42 | 65.15 |
| FedAdagrad | 1×1×1 | | 67.35 | 67.66 | 67.88 | | 68.25 | 68.53 | 69.10 |
| FedAvg | 1×1×1 | | 66.97 | 67.49 | 66.17 | | 68.51 | 67.42 | 65.15 |
| Mean | 1×1×1 | | 66.09 | 68.06 | 68.14 | | 68.26 | 67.20 | 68.64 |
| Ensemble | 1×1×1 | | 68.03 | 65.90 | 67.98 | | 69.07 | 66.57 | 68.64 |
| Combine | 1×1×1 | | 67.58 | 66.99 | 66.58 | | 68.51 | 67.42 | 65.15 |
| YOCO$_{Init\&B\bar{A}}$ | 1×1×1 | | **69.96** | **69.10** | **68.49** | | **69.19** | **68.85** | **70.11** |

**Different MLLMs and client numbers.** We evaluate the impact of two MLLM versions with $N = 10, 20,$ and $30$ clients, as shown in Table 4. Key findings: 1) YOCO consistently achieves top one-shot performance, with improvements of up to 6.14% over Local baseline and up to 4.96% over other baselines. 2) It is more robust than multi-round methods, maintaining comparable or superior performance, particularly at $N = 20$ and $30$, even under low communication conditions.

**Different LoRA ranks and components.** To investigate the effect of rank on performance, we evaluate configurations with ranks $r = 2, 4, 8,$ and $16$, as shown in Figure 7. Performance differences among ranks $4, 8,$ and $16$ are marginal. Considering the trade-off between performance and parameter efficiency, we adopt rank $8$ as the default. Furthermore, in addition to the default use of the $q$

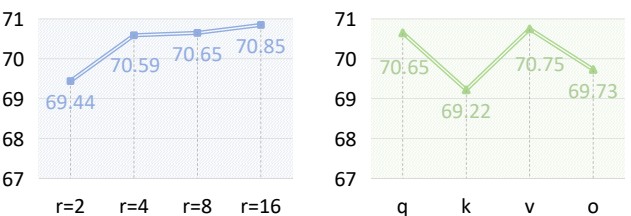

Figure 7: Effect of LoRA ranks $r$=2, 4, 8, 16 and components $q, k, v, o$. $q$ LoRA ($r$=8) balances performance and efficiency.

LoRA component, we explore the $v$, $k$, and $o$ components as well. Results show that the commonly used $q$ and $v$ components yield comparable performance, while $k$ and $o$ lead to slight degradation.

**Cosine similarity matrices.** As shown in the cosine similarity matrix of Figure 8, we compare the LoRAs trained by each client in YOCO$_{Init\&BA}$ with those from other clients, as well as with the aggregated global LoRAs in YOCO$_{Init\&BA}$ and YOCO$_{Init\&B\bar{A}}$. The cosine similarity matrix for LoRA A exhibits lower similarity compared to LoRA B, particularly with YOCO$_{Init\&B\bar{A}}$, which employs Principal Component-weighted Aggregation to enhance the integration of specific knowledge and strengthen adaptability.

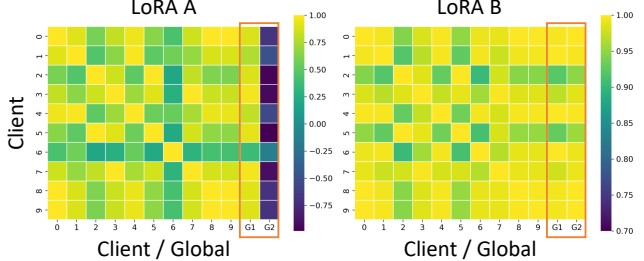

Figure 8: Cosine similarity matrix for LoRA A and B. The $10 \times 10$ matrix shows client-client similarities, while $10 \times 2$ matrix (orange box) shows client-global similarities (G1: YOCO$_{Init\&BA}$, G2: YOCO$_{Init\&B\bar{A}}$). LoRA A has lower cosine similarity, with YOCO$_{Init\&B\bar{A}}$ preserving key specific knowledge.

## 6 Discussion and Conclusion

**Differences from prior works.** Under one-shot federated low-rank adaptation, [36] fine-tunes the model with an additional server-side dataset to extract prototypes for client training. In contrast,

YOCO corrects local bias using implicit global supervision without extra data. To correct local bias, FuseFL [42] aggregates blocks bottom-up, and FENS [3] uses lightweight aggregators, but both violate one-shot communication constraint. YOCO is the first to correct bias via implicit supervision in a true one-shot setting, making a fair performance comparison with FuseFL and FENS infeasible.

**One-shot communication.** This work tackles the challenge of using implicit global supervision to correct local bias and improve performance under one-shot communication. In multi-round scenarios, explicit global supervision is typically gained through model aggregation. Whether implicit supervision provides additional benefits over explicit supervision remains an open question, requiring further empirical investigation. We look forward to future research on this topic.

**Limitations.** YOCO is better suited for LoRA in initialization and regularization, while local bias correction in other PEFT methods remains unresolved. We use the last-layer LoRA for fine-tuning, but this is suboptimal. Future work could explore automated parameter selection for better efficiency.

In conclusion, we propose YOCO, a true one-shot communication method, after analyzing the rationality of implicit global supervision to correct local biases. YOCO introduces sign-based and noise-free regularization strategies, along with a PCA-based aggregation, to enhance consistency and adaptability, building on SVD initialization, which ensures clients share a unified global starting point. Experiments demonstrate the effectiveness of YOCO with minimal communication cost.

## Acknowledgments and Disclosure of Funding

The work is supported by the National Natural Science Foundation of China (Grant No. 62222207, 62427808), the National Research Foundation, Singapore, under its NRF Fellowship (Award# NRF-NRFF14-2022-0001), and the China Scholarship Council program.

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

## Organization of the Appendix

## A  More Related Works

**One-shot federated learning.** In recent years, One-shot Federated Learning (OFL) [32, 4]—requiring only a single round of communication between the server and clients—has attracted increasing research interest due to its significant advantages in communication efficiency and data privacy over traditional Federated Learning (FL). Current approaches to OFL can be broadly classified into four categories: knowledge distillation [40, 15, 64, 13, 21], generative models [64, 13, 21, 59, 58], static ensemble methods [45, 35, 15, 64], and adaptive ensemble methods [13, 42, 3]. Knowledge distillation can be divided into data distillation and model distillation. In data distillation [40, 66], clients generate distilled data from private data and upload it to the server for model training. In model distillation [15, 64, 13, 21], the knowledge from ensembled client models is distilled into the server model using public or synthetic data. Due to the distribution gap between public and private data, some generative model methods [64, 13, 21, 59, 58] for synthetic data have emerged. Generative models are mainly classified into generative adversarial networks (GANs) [64, 13], variational autoencoders (VAEs) [21], and stable diffusion models [59, 58].

Without additional data or support from generative models, static [45, 35] and adaptive [42, 3] ensemble methods offer a direct solution to data and model heterogeneity, while their high flexibility allows seamless integration with other approaches. Early static ensemble methods [16, 45] primarily constructed the global model by applying strategies such as averaging, selection, or output maximization to the models uploaded by clients. Unlike static ensemble methods that rely on extracting fixed characteristics from client models to aid global model construction, recent adaptive ensemble approaches aggregate global supervision developed from the perspectives of causal graphs [42] and secondary generalizers [3]. This aggregated global supervision is used to correct local training bias, thereby achieving strong global model performance. However, aggregated global supervision violates the one-shot communication constraint of OFL, increasing the risk of attacks. To address this, our YOCO employs implicit global supervision—using an initial prior instead of aggregation—to correct local bias while preserving true one-shot communication.

**Federated learning with LoRA.** Most existing federated learning with LoRA approaches [7] adopt multi-round training and primarily focus on addressing three issues: server-side aggregation bias [41, 26, 11, 50, 39, 8], data heterogeneity [17, 20, 33, 9, 60, 56, 28], and computational heterogeneity [12, 6, 5, 31, 50]. To mitigate the issue of server-side aggregation bias, one class of methods [41, 26, 11] adopts an alternating training strategy for the A and B matrices in LoRA, while another explicitly computes the residual between the ideal global LoRA and the aggregated global LoRA [39, 8]. To address data heterogeneity, some methods [17, 20, 33, 9] use two LoRA modules or assign matrices A and B to capture shared and specific knowledge, respectively. Moreover, FRLoRA [56] addresses data heterogeneity and expands the LoRA parameter space by adding the global model to the residual LoRA and reinitializing parameters each local training. To tackle computational heterogeneity, FLoRA [50] and FlexLoRA [6] multiply matrices A and B before aggregation, while HETLORA [12] directly pads lower-rank LoRA modules with zeros. Compared to multi-round FL with LoRA, research on OFL with LoRA under a single communication round is

limited. FedInc [36] fine-tunes LoRA using server-side public data and generates prototypes to assist local training. Unlike FedInc, which relies on additional public data and prototype transmission, our method achieves true one-shot communication in OFL with LoRA by transmitting only the LoRA weights, without any extra data.

## B  Detailed Proof of Lemma 3.1

For a given parameter $i$, let $k_i$ denote the number of client models with a positive sign, and $N - k_i$ denote the number with a negative sign. $N$ is the total number of clients. Then, the total number of client pairs exhibiting sign disagreement on this parameter is $k_i(N - k_i)$. The total number of client pairs is $\binom{N}{2} = \frac{N(N-1)}{2}$, and thus the pairwise conflict rate $\rho_i$ for parameter $i$ is:

$$\rho_i = \frac{2k_i(N - k_i)}{N(N - 1)} \tag{6}$$

The average pairwise conflict rate $\bar{\rho}$ is defined as the mean of the pairwise conflict rates $\rho$ across all parameters $i$:

$$\bar{\rho} = \frac{1}{M} \sum_{i=1}^{M} \rho_i = \frac{2}{MN(N - 1)} \sum_{i=1}^{M} k_i(N - k_i) \tag{7}$$

where $M = \text{card}(\mathbf{\Delta W})$ represents the number of parameters in $\mathbf{\Delta W}$. The overall conflict rate $C$ is defined as:

$$C = \frac{1}{MN} \sum_{i=1}^{M} \min(k_i, N - k_i) \tag{8}$$

By the identity $\min(k_i, N - k_i) = \frac{N - |2k_i - N|}{2}$, $k_i(N - k_i)$ can be expanded as:

$$k_i(N - k_i) = N \cdot \min(k_i, N - k_i) - [\min(k_i, N - k_i)]^2 \tag{9}$$

Substitution into the expression of $\bar{\rho}$ gives:

$$\bar{\rho} = \frac{2}{MN(N - 1)} \left[ N \sum_{i=1}^{M} \min(k_i, N - k_i) - \sum_{i=1}^{M} [\min(k_i, N - k_i)]^2 \right] \tag{10}$$

In conjunction with the definition of $C$:

$$\sum_{i=1}^{M} \min(k_i, N - k_i) = MN \cdot C \tag{11}$$

Therefore, the relationship between the average value $\bar{\rho}$ of the pairwise conflict rate $\rho$ and the overall conflict rate $C$ is given by:

$$\bar{\rho} = \frac{2N}{N - 1} \left[ C - \frac{1}{MN^2} \sum_{i=1}^{M} [\min(k_i, N - k_i)]^2 \right] \tag{12}$$

$\bar{\rho}$ is positively correlated with $C$, but the exact mathematical form depends on the distribution of parameter conflicts.

**IID setting.** For the sake of simplifying the proof, the global LoRA weights $\mathbf{\Delta W_g}$ are computed by averaging the weights from all clients:

$$\mathbf{\Delta W_g} = \frac{1}{N} \sum_{n=1}^{N} \mathbf{\Delta W}_n \tag{13}$$

Let each parameter $\mathbf{\Delta W}_{ni}$ have sign $s_{ni} = \text{sign}(\mathbf{\Delta W}_{ni})$ and magnitude $a_{ni} = |\mathbf{\Delta W}_{ni}|$, where $a_{ni}$ follows a distribution with mean $\mu$ and variance $\sigma^2$. Given parameter $i$ with an overall conflict rate $C_i = \frac{\min(k_i, N - k_i)}{N}$, if the majority of signs are positive (probability $1 - C_i$), then $s_i = +1$; otherwise, if the majority are negative (probability $C_i$), then $s_i = -1$. The expectation of parameter $i$ is expressed as:

$$\mathbb{E}[s_i a_i] = (1 - C_i) \cdot \mu + C_i \cdot (-\mu) = (1 - 2C_i)\mu \tag{14}$$

The expected value of the global weight $\boldsymbol{\Delta W_g}$ is the average of the expected values of all parameters:

$$\mathbb{E}[\boldsymbol{\Delta W_g}] = \frac{1}{M} \sum_{i=1}^{M} \mathbb{E}[s_i a_i] = \frac{1}{M} \sum_{i=1}^{M} (1 - 2C_i)\mu \tag{15}$$

Upon substitution of $C = \frac{1}{MN} \sum_{i=1}^{M} \min(k_i, N - k_i) = \frac{1}{M} \sum_{i=1}^{M} C_i$, we obtain:

$$\mathbb{E}[\boldsymbol{\Delta W_g}] = \mu \left( 1 - 2 \cdot \frac{1}{M} \sum_{i=1}^{M} C_i \right) = (1 - 2C)\mu \tag{16}$$

The variance of parameter $i$ is:

$$\begin{aligned}
\mathrm{Var}(s_i a_i) &= \mathbb{E}[(s_i a_i)^2] - (\mathbb{E}[s_i a_i])^2 \\
\mathbb{E}[(s_i a_i)^2] &= \mathbb{E}[a_i^2] = \sigma^2 + \mu^2; \quad (\mathbb{E}[s_i a_i])^2 = (1 - 2C_i)^2 \mu^2 \\
\mathrm{Var}(s_i a_i) &= \sigma^2 + \mu^2 - (1 - 4C_i + 4C_i^2)\mu^2 = \sigma^2 + 4C_i(1 - C_i)\mu^2
\end{aligned} \tag{17}$$

Since the global weight is the average of the client weights and the parameters are independent, the variance is given by:

$$\mathrm{Var}(\boldsymbol{\Delta W_g}) = \frac{1}{N} \cdot \frac{1}{M} \sum_{i=1}^{M} \mathrm{Var}(s_i a_i) \tag{18}$$

Substituting the variance of a single parameter:

$$\mathrm{Var}(\boldsymbol{\Delta W_g}) = \frac{1}{NM} \sum_{i=1}^{M} [\sigma^2 + 4C_i(1 - C_i)\mu^2] \tag{19}$$

By substituting the overall conflict rate $C = \frac{1}{M} \sum_{i=1}^{M} C_i$ and assuming a uniform distribution of the conflict rates (i.e., $C_i \approx C$), we obtain the approximation:

$$\mathrm{Var}(\boldsymbol{\Delta W_g}) \approx \frac{1}{N} [\sigma^2 + 4C(1 - C)\mu^2] \tag{20}$$

If the overall conflict rates are not uniformly distributed, higher-order correction terms need to be introduced:

$$\mathrm{Var}(\boldsymbol{\Delta W_g}) = \frac{1}{N} [\sigma^2 + 4C(1 - C)\mu^2 - 4\mu^2 \cdot \mathrm{Var}(C_i)] \tag{21}$$

where $\mathrm{Var}(C_i) = \frac{1}{M} \sum_{i=1}^{M} (C_i - C)^2$. For simplicity, we typically assume $\mathrm{Var}(C_i) \approx 0$, implying a uniform distribution of conflict rates. Other patterns warrant future exploration.

**Non-IID setting.** The above derivation of $\mathbb{E}[\boldsymbol{\Delta W_g}]$ and $\mathrm{Var}(\boldsymbol{\Delta W_g})$ assumes that client data is IID. To account for the non-IID setting, we provide a more general derivation below. The main differences lie in modeling parameter heterogeneity across clients—i.e., each client may have different means and variances—and in allowing the conflict rate $C_i$ to vary across parameter dimensions.

Similarly, the global LoRA weights are obtained by averaging the weights from all clients, i.e., $\boldsymbol{\Delta W_g} = \frac{1}{N} \sum_{n=1}^{N} \boldsymbol{\Delta W}_n$. For client $n$, $\boldsymbol{\Delta W}_{ni}$ has sign $s_{ni} = \mathrm{sign}(\boldsymbol{\Delta W}_{ni})$ and magnitude $a_{ni} = |\boldsymbol{\Delta W}_{ni}|$, with $a_{ni}$ following a distribution of mean $\mu_{n,i}$ and variance $\sigma_{n,i}^2$. The overall conflict rate of parameter $i$ is $C_i = \frac{\min(k_i, N - k_i)}{N}$, where $k_i$ is the number of clients with a positive sign. For parameter $i$, the global weight is:

$$\boldsymbol{\Delta W}_{g,i} = \frac{1}{N} \sum_{n=1}^{N} s_{ni} a_{ni} \tag{22}$$

Since the sign $s_{ni}$ has a conflict rate of $C_i$, and the mean magnitude for each client is $\mu_{n,i}$, the expected value is:

$$\mathbb{E}[s_{ni} a_{ni}] = \begin{cases} \mu_{n,i} & \text{with probability } 1 - C_i(\text{majority sign} = +) \\ -\mu_{n,i} & \text{with probability } C_i(\text{majority sign} = -) \end{cases} \tag{23}$$

Therefore, the global expectation is:

$$\mathbb{E}[\mathbf{\Delta W}_{g,i}] = \frac{1}{N}\sum_{n=1}^{N}\mathbb{E}[s_{ni}a_{ni}] = \frac{1}{N}\sum_{n=1}^{N}(1-2C_i)\mu_{n,i} \tag{24}$$

Let the global mean be defined as $\mu_i = \frac{1}{N}\sum_{n=1}^{N}\mu_{n,i}$, then:

$$\mathbb{E}[\mathbf{\Delta W}_{g,i}] = (1-2C_i)\mu_i \tag{25}$$

Compute the mean over all $M$ parameters:

$$\mathbb{E}[\mathbf{\Delta W}_g] = \frac{1}{M}\mathbb{E}[\mathbf{\Delta W}_{g,i}] = \frac{1}{M}\sum_{i=1}^{M}(1-2C_i)\mu_i \tag{26}$$

Defining the overall conflict rate as $C = \frac{1}{M}\sum_{i=1}^{M}C_i$ and the global average magnitude as $\mu = \frac{1}{M}\sum_{i=1}^{M}\mu_i$, the global expectation can then be decomposed as:

$$\mathbb{E}[\mathbf{\Delta W_g}] = \mu - 2\cdot\frac{1}{M}\sum_{i=1}^{M}C_i\mu_i \tag{27}$$

Assuming that $C_i$ and $\mu_i$ are independent, i.e., the conflict rate is unrelated to the parameter magnitude, other cases are left for future work. Then:

$$\mathbb{E}[\mathbf{\Delta W_g}] \approx \mu - 2C\mu = (1-2C)\mu \tag{28}$$

The variance of parameter $i$ can be decomposed into:

$$\mathrm{Var}(\mathbf{\Delta W}_{g,i}) = \mathrm{Var}\left(\frac{1}{N}\sum_{n=1}^{N}s_{ni}a_{ni}\right) \tag{29}$$

Under the assumption that clients are independent, i.e., $\mathrm{Cov}(s_{ni}a_{ni}, s_{mi}a_{mi}) = 0$ for $n \neq m$, which is a common simplification in the literature, especially for theoretical analysis, the variance of parameter $i$ can be decomposed as:

$$\mathrm{Var}(\mathbf{\Delta W}_{g,i}) = \frac{1}{N^2}\sum_{n=1}^{N}\mathrm{Var}(s_{ni}a_{ni}) \tag{30}$$

However, in non-IID settings, this independence assumption may not strictly hold, and cross-client co-variances could potentially affect the global variance. We leave the investigation of these correlations for future work. The variance of the parameters for each client is:

$$\mathrm{Var}(s_{ni}a_{ni}) = \mathbb{E}[(s_{ni}a_{ni})^2] - (\mathbb{E}[s_{ni}a_{ni}])^2$$
$$\mathbb{E}[(s_{ni}a_{ni})^2] = \mathbb{E}[a_{ni}^2] = \sigma_{n,i}^2 + \mu_{n,i}^2; \quad (\mathbb{E}[s_{ni}a_{ni}])^2 = (1-2C_i)^2\mu_{n,i}^2 \tag{31}$$
$$\mathrm{Var}(s_{ni}a_{ni}) = \sigma_{n,i}^2 + \mu_{n,i}^2 - (1-4C_i+4C_i^2)\mu_{n,i}^2 = \sigma_{n,i}^2 + 4C_i(1-C_i)\mu_{n,i}^2$$

By averaging the variances of all clients, we obtain:

$$\mathrm{Var}(\mathbf{\Delta W}_{g,i}) = \frac{1}{N^2}\sum_{n=1}^{N}[\sigma_{n,i}^2 + 4C_i(1-C_i)\mu_{n,i}^2] \tag{32}$$

Further decomposed into two parts: the average variance within clients $\sigma_i^2 = \frac{1}{N}\sum_{n=1}^{N}\sigma_{n,i}^2$, and the variance of mean differences between clients $d_i^2 = \frac{1}{N}\sum_{n=1}^{N}(\mu_{n,i}-\mu_i)^2$. By expanding $\mu_{n,i}^2 = (\mu_i + (\mu_{n,i}-\mu_i))^2$, we obtain:

$$\frac{1}{N}\sum_{n=1}^{N}\mu_{n,i}^2 = \mu_i^2 + d_i^2 \tag{33}$$

Thus, the variance of parameter $i$ is:

$$\mathrm{Var}(\mathbf{\Delta W}_{g,i}) = \frac{1}{N}[\sigma_i^2 + 4C_i(1-C_i)(\mu_i^2 + d_i^2)] \tag{34}$$

Take the average over all $M$ parameters:

$$\text{Var}(\mathbf{\Delta W_g}) = \frac{1}{M}\sum_{i=1}^{M}\text{Var}(\mathbf{\Delta W}_{g,i}) = \frac{1}{NM}\sum_{i=1}^{M}[\sigma_i^2 + 4C_i(1-C_i)(\mu_i^2 + d_i^2)] \qquad (35)$$

If the overall conflict rates $C_i \approx C$, and the means and variances are similar across clients $\mu_i \approx \mu$, $\sigma_i^2 \approx \sigma^2$, $d_i^2 \approx d^2$, then the global variance can be approximated as:

$$\text{Var}(\mathbf{\Delta W}_g) \approx \frac{1}{N}[\sigma^2 + d^2 + 4C(1-C)\mu^2] \qquad (36)$$

Assuming sign conflicts only affect the global expected direction, while client magnitude differences $(d^2)$ exist independently. Thus, $d^2$ is retained as a direct measure of data heterogeneity. Specifically, $d^2$ captures magnitude distribution differences, and $C$ reflects sign conflict levels. Since they are mathematically uncorrelated and physically independent, they can be approximated as separate additive terms in the model. If the overall conflict rate $C_i$ is unevenly distributed, higher-order terms need to be introduced:

$$\text{Var}(\mathbf{\Delta W_g}) = \frac{1}{N}[\sigma^2 + d^2 + 4C(1-C)\mu^2 - 4\mu^2 \cdot \text{Var}(C_i)] \qquad (37)$$

where $\text{Var}(C_i) = \frac{1}{M}\sum_{i=1}^{M}(C_i - C)^2$. The above derivation results are summarized as follows:

IID setting:

$$\begin{cases} \mathbb{E}[\mathbf{\Delta W_g}] = (1-2C)\mu \\ \text{Var}(\mathbf{\Delta W_g}) \approx \frac{1}{N}[\sigma^2 + 4C(1-C)\mu^2] \end{cases}$$

Non-IID setting:

$$\begin{cases} \mathbb{E}[\mathbf{\Delta W_g}] \approx (1-2C)\mu \\ \text{Var}(\mathbf{\Delta W_g}) \approx \frac{1}{N}[\sigma^2 + d^2 + 4C(1-C)\mu^2] \end{cases} \qquad (38)$$

Regardless of whether under the IID or Non-IID setting, the conclusion of Lemma 3.1 holds: reducing the average conflict rate $\bar{\rho}$, decreasing the conflict degree $C$, increasing the expected magnitude of the global update $|\mathbb{E}[\mathbf{\Delta W_g}]|$, and reducing its variance $\text{Var}(\mathbf{\Delta W_g})$ all contribute to stabilizing the global update direction, thereby effectively mitigating local training bias.

## C  Theoretical Guarantees for Convergence and Optimality

Let $W$ denote all trainable LoRA parameters, while the pretrained backbone remains frozen. We analyze single-round ($T = 1$) federated optimization under both IID and non-IID data, explicitly modelling sign conflicts among clients. Two new hypotheses are added to the classical smooth-convex framework to capture inter-client directional disagreement.

**Assumptions**

Let the global objective be $\mathcal{L}(W)$. We assume:

- **(A1) L-smoothness:**
$$\|\nabla\mathcal{L}(W) - \nabla\mathcal{L}(W')\| \leq L\|W - W'\|$$

- **(A2) $\lambda$-strong convexity:**
$$\langle \nabla\mathcal{L}(W) - \nabla\mathcal{L}(W'), W - W' \rangle \geq \lambda\|W - W'\|^2$$

- **(A3) Bounded per-sample gradient variance:**
$$\text{Var}_{(x,y)\sim D_j}[\nabla\ell(f_W(x), y)] \leq \sigma_L^2$$

- **(A4) Bounded sign-conflict variance:**
$$\frac{1}{M}\sum_{i=1}^{M}(C_i - C)^2 \leq \zeta^2$$

- **(A5) Gradient-sign alignment:**
$$\mathbb{E}[\langle \nabla\mathcal{L}_n(W), \nabla\mathcal{L}(W)\rangle] \geq \gamma(1-2C)\|\nabla\mathcal{L}(W)\|^2$$
  where $C = \frac{1}{M}\sum_{i=1}^{M}C_i$, $C \leq 0.5$, and $\gamma \in (0, 1]$ quantifies directional coherence.

## Convergence under IID Data

From Lemma 3.1, the global update across $N$ clients satisfies:

$$\mathbb{E}[\Delta W_g] = (1 - 2C)\mu, \qquad \mathrm{Var}(\Delta W_g) \approx \frac{1}{N}[\sigma^2 + 4C(1 - C)\mu^2]. \tag{1.1}$$

where $\mu$ denotes the average local update direction and $C$ captures the extent of directional disagreement.

To account for bias arising from such conflicts, we prove the following:

**Lemma C.1** (Gradient Bias Bound). *Under (A1)-(A5), the bias of the aggregated gradient satisfies:*

$$\left\| \mathbb{E}[\nabla \tilde{\mathcal{L}}(W)] - \nabla \mathcal{L}(W) \right\| \leq 2\zeta\mu + \mathcal{O}\left( \sqrt{\frac{\sigma^2 + 4C(1-C)\mu^2}{N}} \right).$$

This result reflects that dispersion in sign conflicts ($\zeta$) introduces a systematic bias, while the variance term scales inversely with client number $N$.

**Theorem C.1** (IID Convergence). *Under (A1)-(A5), with $\eta = \frac{1}{L + \lambda^{-1}K}$:*

$$\mathbb{E}[\mathcal{L}(W_1) - \mathcal{L}(W^*)] \leq (1 - \eta\lambda)[\mathcal{L}(W_0) - \mathcal{L}(W^*)] + \frac{\sigma_{\text{eff}}^2}{2\lambda N},$$

*where*

$$\sigma_{\text{eff}}^2 = \sigma^2 + 4C(1 - C)\mu^2 + 4\zeta^2\mu^2, \quad K = \frac{4C(1 - C)\mu^2}{N}.$$

The term $\left(1 - \frac{\lambda}{L + \lambda^{-1}K}\right)$ captures the contraction due to strong convexity and smoothness, degraded by conflict-induced variance $K$. The residual error $\frac{\sigma_{\text{eff}}^2}{2\lambda N}$ includes noise from client sampling ($\sigma^2$), sign conflict ($C$), and dispersion ($\zeta$).

## Extension to Non-IID Settings

Under data heterogeneity, we define $d^2$ as the inter-client drift (see Appendix B):

$$d^2 = \frac{1}{M} \sum_{i=1}^{M} \frac{1}{N} \sum_{n=1}^{N} (\mu_{n,i} - \mu_i)^2, \tag{1.2}$$

measuring deviation in client update directions.

The global update variance now becomes:

$$\mathrm{Var}(\Delta W_g) \approx \frac{1}{N}[\sigma^2 + d^2 + 4C(1 - C)\mu^2]. \tag{1.3}$$

**Theorem C.2** (Non-IID Convergence). *Under data heterogeneity and (A1)-(A5), with $\eta = \frac{1}{L + \lambda^{-1}K_{niid}}$:*

$$\mathbb{E}[\mathcal{L}(W_1) - \mathcal{L}(W^*)] \leq (1 - \eta\lambda\gamma(1 - 2C)^2)[\mathcal{L}(W_0) - \mathcal{L}(W^*)] + \frac{\sigma_{niid}^2}{2\lambda N},$$

*where*

$$K_{niid} = \frac{4C(1 - C)\mu^2 + d^2}{N}, \quad \sigma_{niid}^2 = \sigma^2 + d^2 + 4C(1 - C)\mu^2 + 4\zeta^2\mu^2.$$

The exponential term $\exp\left(-\frac{\lambda\gamma(1-2C)^2}{L + \lambda^{-1}K_{niid}}\right)$ encodes alignment efficiency $\gamma$ and conflict rate $C$, attenuated by drift $d^2$. The residual error $\frac{\sigma_{niid}^2}{2\lambda N}$ aggregates variances from sampling, conflict, dispersion, and heterogeneity.

**Asymptotic Optimality (One-shot Limit)**

The residual error as $N \to \infty$ (large client count) simplifies to:

$$\lim_{N \to \infty} \mathbb{E}[\mathcal{L}(W_1) - \mathcal{L}(W^*)] \leq \frac{\sigma_{\text{niid}}^2}{2\lambda N} \approx \frac{d^2 + 4\zeta^2 \mu^2}{2\lambda N}. \tag{1.4}$$

This confirms that conflict ($C$) and dispersion ($\zeta$) remain additive error sources even in the one-shot setting, alongside data heterogeneity ($d^2$).

**Key Implications for $T = 1$**

1. **Conflict Sensitivity:** Optimality gap scales with $C(1 - C)$ and $\zeta^2$, peaking at $C = 0.5$. Alignment strength $\gamma$ critically amplifies the exponential decay in non-IID settings.

2. **Client Scaling:** Residual error decays as $\mathcal{O}(1/N)$, emphasizing the benefit of large client pools to mitigate variance.

3. **Algorithmic Guidance:** Conflict-aware aggregation (e.g., reweighting clients with low $C_i$) directly reduces $\sigma_{\text{eff}}^2$ and $\sigma_{\text{niid}}^2$. Drift reduction techniques (e.g., regularization) remain essential for non-IID robustness.

These results provide theoretical grounding for federated LoRA fine-tuning in communication-constrained scenarios, highlighting the irreducible impact of client disagreement even in single-round optimization.

# D Scalability Evaluation Beyond the Benchmark Setting

Based on the adopted benchmark, we primarily evaluated YOCO with 10-30 clients. The method, however, is not restricted to this range: constrained only by GPU resources, we further conducted experiments with 50 clients and analyzed the scalability of the conflict-matrix computation. As shown in Table 5, YOCO consistently maintains its effectiveness at larger scales, outperforming both basic one-shot baselines and multi-round FL approaches.

Table 5: Performance comparison with $N = 50$ clients under two MLLM variants: V2_6 (MiniCPM-V-2_6int4) and V2_5 (MiniCPM-Llama3-V-2_5int4). Comm. cost denotes communication overhead.

| Method | Comm. cost | $N = 50$ (**V2_6**) | $N = 50$ (**V2_5**) |
|---|---|---|---|
| FedAdam | 25×28×4 | 66.67 | 68.64 |
| Local | 0×1×1 | 65.71 | 62.93 |
| FedAdam | 1×1×1 | 67.69 | 69.09 |
| FedAvgM | 1×1×1 | 66.03 | 67.69 |
| FedAdagrad | 1×1×1 | 67.07 | 69.18 |
| FedAvg | 1×1×1 | 66.23 | 67.69 |
| Mean | 1×1×1 | 67.58 | 68.31 |
| Combine | 1×1×1 | 65.84 | 68.09 |
| YOCO$_{\text{initB}\bar{\text{A}}}$ | 1×1×1 | 68.15 | 69.83 |

**IID Scenario: Statistical-Scale Derivation as $N \to \infty$**

Let the probability that parameter $i$ takes a positive sign be $p_i$. Then

$$k_i \sim \text{Binom}(N, p_i), \qquad 0 \leq p_i \leq 1.$$

**Expected Global Conflict Rate**

$$\mathbb{E}[C] = \frac{1}{MN} \sum_i \mathbb{E}[\min(k_i, N - k_i)] \approx \frac{1}{M} \sum_i \left\{ \frac{N}{2} \left[ 1 - 2|p_i - \tfrac{1}{2}| \right] - O(\sqrt{N}) \right\} \Big/ N.$$

- If $p_i \approx 0.5$ (signs nearly balanced), $\mathbb{E}[C] \to \frac{1}{2} - \Theta(N^{-1/2})$.
- If $p_i$ is biased to one side, then $\mathbb{E}[C] \ll 0.5$ and decreases further with $N$.

**Conclusion 1.** In non-extreme mixed-sign settings, $C$ **does not grow with** $N$; indeed, because most parameters converge to the same sign, its upper bound decreases monotonically as $N$ increases.

**Global Weight Noise**

The original derivation gives

$$\mathrm{Var}(\Delta W_g) = \frac{1}{N}\big[\sigma^2 + 4C(1-C)\mu^2\big].$$

- Leading term $\sigma^2/N \Rightarrow$ **variance** $\propto 1/N$ — more clients make each global update more stable.
- Secondary term $4C(1-C)\mu^2 \Rightarrow$ shrinks as $C$ decreases; YOCO's directional constraint is designed to reduce $C$.

**Conclusion 2.** With hundreds or thousands of clients, the $O(1/N)$ noise is already far smaller than directional bias, so YOCO still delivers significant gains.

**Non-IID Extension: Cluster-Level Beta–Binomial Model**

Partition the clients into $G$ clusters, with proportions $\alpha_g$ ($\sum_g \alpha_g = 1$). Within cluster $g$, the positive-sign probability of parameter $i$ is $p_i^{(g)}$:

$$k_i^{(g)} \sim \mathrm{Binom}(\alpha_g N, \, p_i^{(g)}), \qquad k_i = \sum_{g=1}^{G} k_i^{(g)}.$$

**Conflict Rate**

$$\mathbb{E}[C] = \frac{1}{MN}\sum_i \sum_g \mathbb{E}\big[\min(k_i^{(g)}, \alpha_g N - k_i^{(g)})\big] \; + \; O(G^2) = O\big(\tfrac{1}{N}\big) + O\big(\tfrac{G^2}{N^2}\big).$$

When $G \ll N$ (typical in practice), the dominant term remains $O(1/N)$.

**Variance**

$$\mathrm{Var}(\Delta W_g) = \frac{1}{N}\Big[\sigma^2 + 4C(1-C)\mu^2 + O\big(\tfrac{G}{N}\big)\Big],$$

which differs from the IID result only by the $O(G/N)$ adjustment.

**Conclusion 3.** As long as each cluster contains $\Omega(N)$ clients (equivalently, $G \ll N$), the $O(1/N)$ scaling of both $C$ and $\mathrm{Var}(\Delta W_g)$ is retained, and YOCO's directional-stability effect remains valid.

# E Details of Experiment Configuration

## E.1 Hyper Parameters

As shown in Table 6, we provide the specific hyperparameter settings of $\gamma$ and $\lambda$ for sign-based consistency regularization and noise-free sparse regularization, respectively.

## E.2 Aggregation Strategies

As shown in Tables 7 and 8, the aggregation strategies of different YOCO versions are presented.

# F Details of Algorithm

Details of our algorithm is shown in Algorithm 1.

Table 6: Hyperparameter settings for $\gamma$ and $\lambda$.

| Hateful-Memes | $\alpha$=5.0 | $\alpha$=1.0 | $\alpha$=0.5 | $\beta$=30% | $\beta$=40% | $\beta$=50% |
|---|---|---|---|---|---|---|
| $\gamma$ | $(1, 50)$ | $(1, 25)$ | $(1, 200)$ | $(1, 50)$ | $(1, 100)$ | $(1, 100)$ |
| $\lambda$ | $1e{-}2$ | $1e{-}1$ | $1e{-}4$ | $1e{-}2$ | $1e{-}2$ | $1e{-}3$ |
| Hateful-Memes | I:3-T:7 | I:5-T:5 | I:7-T:3 | $p$=80% | $p$=70% | $p$=60% |
| $\gamma$ | $(1, 300)$ | $(1, 100)$ | $(1, 25)$ | $(1, 100)$ | $(1, 25)$ | $(1, 100)$ |
| $\lambda$ | $1e{-}3$ | $1e{-}2$ | $1e{-}2$ | $1e{-}1$ | $1e{-}3$ | $1e{-}4$ |
| CrisisMMD | $\alpha$=5.0 | $\alpha$=1.0 | $\alpha$=0.5 | $\beta$=30% | $\beta$=40% | $\beta$=50% |
| $\gamma$ | $(1, 50)$ | $(1, 5)$ | $(1, 200)$ | $(1, 50)$ | $(1, 100)$ | $(1, 25)$ |
| $\lambda$ | $1e{-}4$ | $1e{-}3$ | $1e{-}2$ | $1e{-}4$ | $1e{-}3$ | $1e{-}4$ |
| CrisisMMD | I:3-T:7 | I:5-T:5 | I:7-T:3 | $p$=80% | $p$=70% | $p$=60% |
| $\gamma$ | $(1, 5)$ | $(1, 25)$ | $(1, 25)$ | $(1, 50)$ | $(1, 50)$ | $(1, 25)$ |
| $\lambda$ | $1e{-}4$ | $1e{-}3$ | $1e{-}2$ | $1e{-}4$ | $1e{-}4$ | $1e{-}4$ |
| Medical Datasets | I:3-T:7 | I:5-T:5 | I:7-T:3 | $p$=80% | $p$=70% | $p$=60% |
| $\gamma$ | $(1, 25)$ | $(1, 25)$ | $(1, 25)$ | $(1, 25)$ | $(1, 25)$ | $(1, 25)$ |
| $\lambda$ | $1e{-}4$ | $1e{-}4$ | $1e{-}4$ | $1e{-}4$ | $1e{-}4$ | $1e{-}4$ |

Table 7: Aggregation strategies in YOCO on the Hateful-Memes dataset. "Combine" denotes "FedAvgM" aggregation for LoRA A and "Mean" aggregation for LoRA B.

| Hateful-Memes | $\alpha$=5.0 | $\alpha$=1.0 | $\alpha$=0.5 | $\beta$=30% | $\beta$=40% | $\beta$=50% |
|---|---|---|---|---|---|---|
| YOCO$_{\text{init}}$ | Combine | FedAvg | Combine | Combine | Combine | FedAvg |
| YOCO$_{\text{Init\&B}}$ | FedAvg | FedAvg | Combine | FedAvg | Combine | FedAvg |
| YOCO$_{\text{Init\&BA}}$ | FedAvg | Combine | Combine | FedAvg | FedAvg | FedAvg |
| YOCO$_{\text{Init\&B}\bar{\text{A}}}$ (for B) | FedAvg | FedAvgM | FedAvgM | FedAvg | FedAvg | FedAvg |
| Hateful-Memes | I:3-T:7 | I:5-T:5 | I:7-T:3 | $p$=80% | $p$=70% | $p$=60% |
| YOCO$_{\text{init}}$ | Ensemble | FedAvgM | FedAvgM | Ensemble | FedAdagrad | Ensemble |
| YOCO$_{\text{Init\&B}}$ | Ensemble | FedAvgM | FedAvg | Ensemble | FedAdagrad | Mean |
| YOCO$_{\text{Init\&BA}}$ | Ensemble | FedAvgM | FedAvgM | Ensemble | Ensemble | Mean |
| YOCO$_{\text{Init\&B}\bar{\text{A}}}$ (for B) | FedAvgM | FedAvgM | FedAvgM | FedAdagrad | Mean | FedAvgM |

# G  Experiments on the Single-Modality Setting

To further evaluate the generalizability of YOCO in a single-modality setting, we conducted additional experiments, as presented in Table 9. The results show that YOCO consistently outperforms all one-shot baselines in the single-modality setting and even surpasses multi-round FL on the text modality of CrisisMMD. These findings demonstrate that YOCO remains highly effective in single-modality scenarios, underscoring its strong generalization capability.

# H  More Visualizations of the Conflict Matrix

To illustrate the changes in conflicts on LoRA A and B from YOCO$_{\text{init}}$ to YOCO$_{\text{init\&B}}$, we visualize the conflict matrices under two data heterogeneity settings. As shown in Figures 9 and 10, introducing sign-based consistency regularization for LoRA B effectively reduces its conflicts, while the conflicts on LoRA A remain nearly unchanged.

# I  Broader Impact

Federated learning (FL) is a distributed training framework designed to protect data privacy, making it particularly well-suited for privacy-sensitive domains such as finance and healthcare. In recent years, many multimodal large models (MLLMs) have become increasingly mature, exhibiting strong

Table 8: Aggregation strategies in YOCO on the CrisisMMD and Medical datasets. "Combine" denotes "Mean" aggregation for LoRA A and "FedAvgM" aggregation for LoRA B.

| CrisisMMD | $\alpha$=5.0 | $\alpha$=1.0 | $\alpha$=0.5 | $\beta$=30% | $\beta$=40% | $\beta$=50% |
|---|---|---|---|---|---|---|
| YOCO$_\text{init}$ | Combine | Mean | Combine | Mean | Combine | Combine |
| YOCO$_\text{Init\&B}$ | Combine | Mean | Combine | Mean | Mean | Mean |
| YOCO$_\text{Init\&BA}$ | Mean | Mean | Mean | Combine | Combine | Combine |
| YOCO$_\text{Init\&B\bar{A}}$ (for B) | Mean | Mean | Mean | Mean | Mean | Mean |
| CrisisMMD | I:3-T:7 | I:5-T:5 | I:7-T:3 | $p$=80% | $p$=70% | $p$=60% |
| YOCO$_\text{init}$ | Ensemble | Ensemble | Ensemble | Combine | Ensemble | Ensemble |
| YOCO$_\text{Init\&B}$ | Ensemble | Ensemble | Ensemble | Combine | Ensemble | Ensemble |
| YOCO$_\text{Init\&BA}$ | Ensemble | Ensemble | Ensemble | Mean | Ensemble | Ensemble |
| YOCO$_\text{Init\&B\bar{A}}$ (for B) | Mean | Mean | Mean | FedAdagrad | Mean | Mean |
| Medical Datasets | I:3-T:7 | I:5-T:5 | I:7-T:3 | $p$=80% | $p$=70% | $p$=60% |
| YOCO$_\text{Init\&B\bar{A}}$ (for B) | FedAvgM | FedAvgM | FedYogi | FedYogi | FedYogi | FedYogi |

---

**Algorithm 1** YOCO

1: **Input:** Local dataset $\mathcal{D}_n$, hyperparameters $\gamma, \lambda$, client set $\mathcal{N}$, task loss $\mathcal{L}_s$
2: **Output:** Global weights $\mathbf{\Delta W_g}$
3: Server executes Eq. (3) to obtain $\mathbf{A}^0$ and $\mathbf{B}^0$.                    ▷ **Initialization**
4: **for** each client $n \in \mathcal{N}$ **do**                    ▷ **Local Training with Regularization**
5:        Initialize local LoRA: $\mathbf{A}_n^{(0)} \leftarrow \mathbf{A}^0$; $\mathbf{B}_n^{(0)} \leftarrow \mathbf{B}^0$
6:        $\mathbf{\Delta W}_n = \mathbf{B}_n \mathbf{A}_n$
7:        $\mathcal{R}_\text{sign} = \frac{\|\tanh(\gamma \mathbf{B}_n) - \mathbf{B}^s\|_2}{\|\mathbf{B}^s\|_2 + \varepsilon}$                    ▷ **Sign-based Consistency Regularization**
8:        $\mathcal{R}_\text{sparse} = \lambda \times \|\mathbf{A}_n\|_1$                    ▷ **Noise-free Spare Regularization**
9:        $\mathcal{L}(\mathbf{\Delta W}_n; \mathcal{D}_n) = \mathcal{L}_s + \mathcal{R}_\text{sign} + \mathcal{R}_\text{sparse}$
10:        Update parameters: $\mathbf{\Delta W}_n \leftarrow \mathbf{\Delta W}_n - \eta \nabla \mathcal{L}$
11: **end for**
12: Server Aggregation:                    ▷ **Principal Component-weighted Aggregation**
13: Aggregate $\mathbf{B}_g$ with vanilla strategy, $\mathbf{A}_g$ with Eq. (5)
14: Return global weights $\mathbf{\Delta W_g}$ for evaluation

---

Table 9: Performance comparison under $\alpha = 0.5$ in the single-modality setting. "Comm. cost" denotes communication cost.

| Method | Comm. cost | Hateful-Memes | | Comm. cost | CrisisMMD | |
|---|---|---|---|---|---|---|
| | | Image | Text | | Image | Text |
| FedAdam | 25×28×4 | 72.70 | 68.26 | 40×28×4 | 59.59 | 37.95 |
| Local | 0×1×1 | 67.47 | 65.41 | 0×1×1 | 41.76 | 33.05 |
| FedAdam | 1×1×1 | 67.67 | 66.24 | 1×1×1 | 39.24 | 30.04 |
| FedAvgM | 1×1×1 | 68.99 | 66.43 | 1×1×1 | 44.98 | 32.95 |
| FedAdagrad | 1×1×1 | 67.68 | 66.36 | 1×1×1 | 30.74 | 26.47 |
| FedAvg | 1×1×1 | 68.79 | 66.42 | 1×1×1 | 44.25 | 33.35 |
| Mean | 1×1×1 | 67.36 | 66.49 | 1×1×1 | 53.35 | 35.86 |
| Ensemble | 1×1×1 | 69.07 | 66.39 | 1×1×1 | 53.21 | 36.54 |
| Combine | 1×1×1 | 69.19 | 66.33 | 1×1×1 | 53.45 | 37.08 |
| YOCO$_\text{init}$ | 1×1×1 | 69.22 | 66.31 | 1×1×1 | 53.30 | 37.66 |
| YOCO$_\text{init\&B}$ | 1×1×1 | 69.53 | 66.41 | 1×1×1 | 53.99 | 37.78 |
| YOCO$_\text{init\&BA}$ | 1×1×1 | 69.88 | 66.93 | 1×1×1 | 54.37 | 38.10 |
| YOCO$_\text{init\&B\bar{A}}$ | 1×1×1 | 70.10 | 67.11 | 1×1×1 | 54.51 | 38.65 |

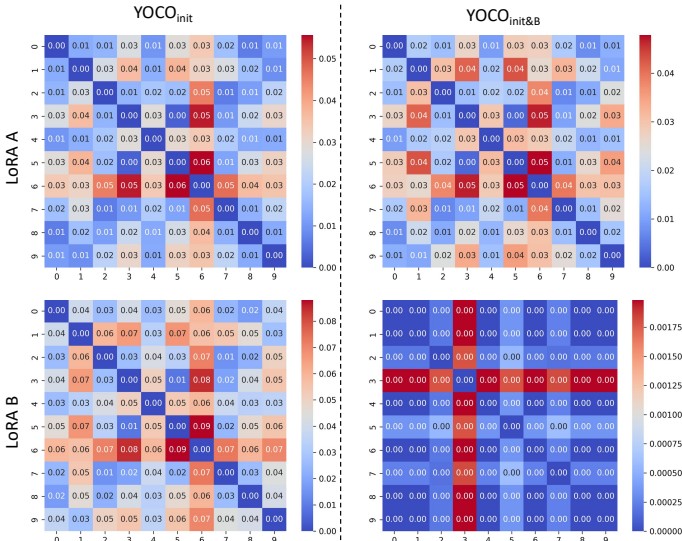

Figure 9: Conflict matrices of YOCO$_{init}$ and YOCO$_{init\&B}$ on Hateful-Memes with $\alpha = 5.0$.

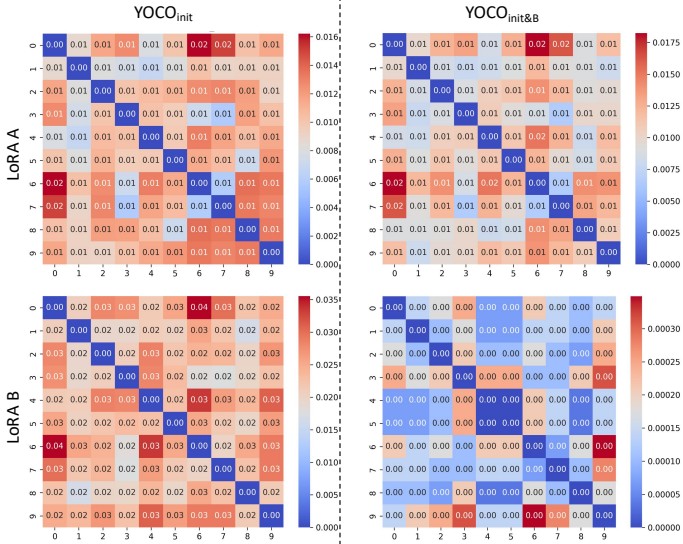

Figure 10: Conflict matrices of YOCO$_{init}$ and YOCO$_{init\&B}$ on Hateful-Memes with $\alpha = 1.0$.

general prior knowledge and the ability to handle a wide range of multimodal tasks in a unified manner. These models can also achieve impressive performance on specific tasks through rapid fine-tuning. Integrating MLLMs with FL offers significant potential to accelerate the development of intelligent systems in privacy-critical settings.

However, conventional MLLMs+FL frameworks typically rely on multiple rounds of communication, making them susceptible to performance degradation in resource-constrained client environments. This limitation poses challenges for deployment on edge devices and in low-resource scenarios. To address this, we propose YOCO, the first framework to enable efficient federated low-rank adaptation of MLLMs under a truly one-shot communication setting. YOCO introduces an implicit global supervision mechanism to mitigate local training bias without relying on additional data or generative models. It substantially reduces communication overhead while maintaining—or even enhancing—model performance.

The introduction of YOCO broadens the applicability of MLLMs+FL, particularly in settings involving resource-limited edge devices, remote healthcare facilities, and small-scale organizations. This advancement holds great promise for promoting the real-world adoption of MLLMs in privacy-sensitive domains such as finance and healthcare.

