# OpenReview forum: "You Only Communicate Once: One-shot Federated Low-Rank Adaptation of MLLM"
_NeurIPS.cc/2025/Conference — NeurIPS 2025 poster_

### Official Review · Reviewer_Cz1u · 2025-06-16

**Clarity:** 2
**Significance:** 2
**Originality:** 3
**Rating:** 2
**Confidence:** 4

**Summary:**

This work introduces YOCO, a true one-shot communication method, after analyzing the rationale behind implicit global supervision for correcting local biases. Specifically, YOCO employs directional supervision with sign-regularized LoRA B to enforce global consistency, while sparsely regularized LoRA A preserves client-specific adaptability. Moreover, it adopts PCA-based aggregation to enhance both consistency and adaptability.

**Questions:**

- I have some doubts about using pre-trained models as “implicit global supervision” to correct local training biases. Since the dataset used for the pre-trained model may differ from the data used in this FL setting, is it really reasonable to use this model as “implicit global supervision” to correct local training biases?
- There is some confusion in Equation (5). For example, what does Z_i represent? How is P obtained?
- Why is LoRA applied only to the last layer's q_proj?

**Ethical Concerns:**

["NO or VERY MINOR ethics concerns only"]

**Final Justification:**

Most of my concerns are not resolved, such as the shared knowledge part, the rationale of the method design of this work, the explanation of the main experimental results, etc.  \
Taking the most important issues, i.e., the method design, as an example: The authors state, "In the visualization of the conflict matrix, where directional constraints were applied simultaneously to both LoRA A and B, we observed that the conflicts in LoRA A increased instead. Based on this observation, we chose to apply global directional constraints only to LoRA B, which originally exhibited larger conflicts, thereby effectively reducing conflict". However, the most important phenomenon, "directional constraints were applied simultaneously to both LoRA A and B, we observed that the conflicts in LoRA A increased instead", is not well explained. Moreover, why apply global directional constraints only to LoRA B and not to A is also not well explained. These experiments (Q10) were added during the rebuttal period, not the method design period, which raises doubts about the rationale of this method design. \
When I continue to ask these questions, the authors seem only to be muddying the issue (by posting 'Main Concern Resolved; New question raised by the reviewer'; however, the main concern remains, and no new question arises) and attacking the reviewer ("We believe such a self-contradictory statement appears irresponsible"), rather than focusing on addressing the reviewer’s concerns. Thus, I stand for the rejection of this work.

**Limitations:**

yes

**Quality:**

2

**Strengths And Weaknesses:**

Strengths:
- This work studies a novel question: using LoRA to fine-tune MLLMs under the OFL setting.
- The design of using pre-trained models as “implicit global supervision” to correct local training biases is novel and interesting.

Weaknesses:
- The paper seems to present conflicting conclusions. Figure 4 indicates that the conflict among LoRA A is smaller than that of LoRA B. Does this mean that A learns shared knowledge (resulting in smaller direct conflicts), while B learns client-specific features (leading to greater conflicts)? Why is it that after PCA in Figure 5, the learned A appears more similar, while B appears less so, leading to the conclusion that "LoRA B learns shared knowledge, while LoRA A captures client-specific features"?
- Some phenomena in this work are not well explained. For example, in lines 132-133, "Constraining both LoRA A and B introduces excessive restrictions, which in turn increases the direction conflicts of LoRA A across clients." Why does this happen?
- In lines 184-185, "The regularization terms encourage LoRA B to learn shared knowledge, while LoRA A captures client-specific features." How is this achieved? Why does the sparse regularization help A learn specific knowledge?
- In Figure 4, what are the results when only using SVD without constraints? Since the effect in Figure 4(d) may be attributed to SVD, it would be better to include results that use only SVD.
- The statement in line 8, "meaning they never attain true one-shot communication," seems a bit too absolute. According to the authors' discussion in Section 2, there are indeed some works that achieve "true one-shot communication."

---

> ### Author Rebuttal · Authors · 2025-07-29
>
> We sincerely appreciate **Reviewer Cz1u** for the insightful comments and constructive questions. We have provided comprehensive responses to clarify these points and look forward to further dialogue and feedback.
>
> >**Q1:** The paper seems to present conflicting conclusions. **(Q1-1)** Figure 4 indicates that the conflict among LoRA A is smaller than that of LoRA B. Does this mean that A learns shared knowledge, while B learns client-specific features? **(Q1-2)** Why is it that after PCA in Figure 5, the learned A appears more similar, while B appears less so, leading to the conclusion that "LoRA B learns shared knowledge, while LoRA A captures client-specific features"?
>
> Thank you for raising these questions. We address Q1-1 and Q1-2 separately in A1-1 and A1-2, providing point-by-point responses.
>
> **A1-1:** We begin by clarifying the definition of **conflict rate**, emphasizing that its value **cannot directly indicate the type of knowledge**. Under unconstrained conditions, the conflict rate is often shaped by multiple factors acting together. We then explain **why LoRA B tends to capture shared knowledge, whereas LoRA A is more inclined to reflect client‑specific knowledge**.
>
> #### **1. Conflict rate $\not\Rightarrow$ Knowledge type**
>
> Let $k_i$ denote the number of client signs consistent on parameter $i$, with $M = \operatorname{card}(\mathbf{\Delta W})$ representing the number of parameters and $N$ the total number of clients. The overall conflict rate $C$ and the average pairwise conflict rate $\bar{\rho}$ are defined as follows (see lines 112-127):
> $$
> C=\frac{1}{MN}\sum_i\min(k_i,N-k_i)
> $$
>
> $$
> \bar{\rho}=\frac{2N}{N-1}\left[C-\frac{1}{MN^2}\sum_i[\min(k_i,N-k_i)]^2\right]
> $$
>
> Both metrics rely only on the multiset ${k_i}_{i=1}^{M}$, failing to capture parameter semantics and being easily affected by initialization, noise, or sign imbalance. Hence, the conflict rate alone cannot determine the type of knowledge learned.
>
> #### **2. Knowledge type of LoRA B and A**
>
> During training, we impose a common **directional constraint** on every client to align the LoRA $B$ matrix with the pre‑trained weight direction. Without this constraint, LoRA updates are dictated entirely by local gradients: the *consistent* component across clients forms shared knowledge, while *differences* constitute specific knowledge.
>
> With the constraint, each client’s LoRA $B$ converges toward the same global direction, explicitly injecting a common component and thus favouring the capture of shared patterns.
> Meanwhile, client individuality is maintained—indeed emphasized—through an $\ell_1$‑norm sparsity regularizer applied to the LoRA $A$ matrix.
>
> **In short:** LoRA B learns more shared knowledge due to explicit global directional guidance, whereas LoRA A, lacking such guidance and relying on local data, becomes increasingly biased toward client‑specific knowledge with more training epochs.
>
> **A1-2:** In the PCA visualisation of **Figure 5**, the features of LoRA $A$ from different clients are **more scattered**, whereas those of LoRA $B$ are **more clustered**. This supports our claim that **LoRA $A$ leans towards client‑specific knowledge, while LoRA $B$ leans towards shared knowledge**.
>
> It is worth noting that in Figure 5, the regions separated by the dashed line correspond to two different data partitioning configurations. The PCA visualization assesses feature similarity based on the degree of dispersion and aggregation: the more aggregated the features, the higher their similarity, and thus the more likely they reflect shared knowledge. Therefore, it is reasonable to conclude that LoRA B tends to capture shared knowledge.
>
> >**Q2:** Some phenomena in this work are not well explained. For example, in lines 132-133, "Constraining both LoRA A and B introduces excessive restrictions, which in turn increases the direction conflicts of LoRA A across clients." Why does this happen?
>
> **A2:** Thank you for the insightful comment. In the OFL setting, imposing identical directional constraints on both LoRA matrices ($A$ and $B$) overly restricts the adaptation space, thereby weakening the model’s ability to handle non‑IID data. Under such restrictions, clients are forced to update within the same narrow subspace. This limitation is particularly severe for the $A$ matrix, which is responsible for core feature projection and is highly sensitive to local data distributions. Consequently, gradient updates on $A$ are more likely to diverge across clients, further amplifying directional inconsistency. We will incorporate this clarification in the final version.
>
> >**Q3:** In lines 184-185, "The regularization terms encourage LoRA B to learn shared knowledge, while LoRA A captures client-specific features." How is this achieved? Why does the sparse regularization help A learn specific knowledge?
>
> **A3:** Thank you for your insightful question. In this study, we impose a unified directional constraint on the LoRA $B$ matrix, aligning it with the direction of the pre-trained weights. This encourages all clients to share a common set of output basis vectors, thereby naturally capturing cross-client shared patterns.
>
> Meanwhile, we apply $\ell_1$‑norm sparsity regularization to the LoRA $A$ matrix, which drives each client to retain only a small number of non-zero coefficients that are most representative of its local data. Due to the heterogeneity in data distributions, the activated sets differ across clients, enabling the model to effectively capture personalized features.
>
> The effectiveness of this design is supported by the feature distribution visualization (Figure 5) and the cosine similarity analysis (Figure 8).
>
> >**Q4:** In Figure 4, what are the results when only using SVD without constraints? Since the effect in Figure 4(d) may be attributed to SVD, it would be better to include results that use only SVD.
>
> **A4:** Thank you for the suggestion. Since figures cannot be included in the rebuttal, we provide a quantitative explanation instead. In the original paper, **YOCO$_{\text{init}}$** (lines 161 and 214) in Tables 1 and 2 corresponds to the **SVD‑only result**, while Figure 4(d) shows YOCO$_{\text{initB}}$, which additionally incorporates the directional constraint on LoRA $B$.
>
> Across most scenarios, YOCO$ _{initB} $ consistently outperforms YOCO$ _{init} $, with the largest gain of **2.23 points** observed at $p=80$% on the CrisisMMD dataset. This confirms that combining SVD with a directional constraint on LoRA $B$ yields better performance than using SVD alone. We will also provide a visualization of the conflict matrix for the SVD‑only case in the final version to further substantiate this conclusion.
>
> >**Q5:** The statement in line 8, "meaning they never attain true one-shot communication," seems a bit too absolute. According to the authors' discussion in Section 2, there are indeed some works that achieve "true one-shot communication."
>
> **A5:** Thank you for pointing this out. The full sentence on line 8 of the original manuscript is:
>
> > *“However, existing adaptive ensemble OFL methods still need more than one round of communication, because correcting heterogeneity‑induced local bias relies on aggregated global supervision, meaning they never attain true one‑shot communication.”*
>
> The conclusion *“they never attain true one‑shot communication”* rests on two premises:
>
> 1. “existing adaptive ensemble OFL methods,” and
> 2. “correcting heterogeneity‑induced local bias relies on aggregated global supervision.”
>
> We sincerely appreciate the reviewer’s attention to the precision of our wording and fully agree with this rigorous academic attitude. In the final version, we will revise the statement to *“meaning they still do not achieve true one‑shot communication”* to convey the intended meaning more accurately.
>
> >**Q6:** I have some doubts about using pre-trained models as “implicit global supervision” to correct local training biases. Since the dataset used for the pre-trained model may differ from the data used in this FL setting, is it really reasonable to use this model as “implicit global supervision” to correct local training biases?
>
> **A6:** Thank you for raising this concern. Due to space limitations, please refer to our response **A2** to **Reviewer P4k5**.
>
> >**Q7:** There is some confusion in Equation (5). For example, what does $Z_i$ represent? How is $P$ obtained?
>
> **A7:** We apologize for the confusion. Here, $Z_i$ denotes the coordinate vector of the $i$‑th client in the top‑$k$ principal components, representing the position of client $i$’s LoRA $A$ in the principal component space.
>
> During PCA, we obtain both the principal component coordinates $Z$ and a loading matrix $P \in \mathbb{R}^{d\times k}$. In the paper, this is written as $P^{\top} \in \mathbb{R}^{k\times d}$—the transpose of the loading matrix—used to reconstruct the original parameter space from the global coordinates $Z_g$.
>
> Here, $Z_g$ is the weighted average of all $Z_i$ vectors, with weights determined by each client’s local data size $|D_i|$. We will add this clarification in the final version to improve clarity and completeness.
>
> >**Q8:** Why is LoRA applied only to the last layer's q_proj?
>
> **A8:** Thank you for raising this point. The choice of LoRA layers—including their number and positions—falls under **parameter selection** for LoRA fine‑tuning, whereas this paper focuses on mitigating **local training bias** under one‑shot communication. While distinct, both are critical for MLLMs with OFL; we leave layer selection strategies to future work.
>
> Our experiments show that fine‑tuning even **one layer** achieves strong performance, sometimes surpassing multi‑round baselines. Evaluations across different single‑layer positions revealed only **minor differences**, with slight variations among $q_{proj}$, $k_{proj}$, $v_{proj}$, and $o_{proj}$, as shown in Figure 7, with $q_{proj}$ and $v_{proj}$ performing slightly better overall.

---

> > ### Comment · Reviewer_Cz1u · 2025-08-03
> >
> > Thanks for the author's response. However, I still have some questions about whether A or B learns shared knowledge.
> >
> > First, what measurement can be used to represent whether it (LoRA A or B) learns shared knowledge or not? Why can the conflict matrices in Figure 4 not represent this, while the PCA visualization can?
> >
> > Second, as the authors claim, "During training, we impose a common directional constraint on every client to align the LoRA $B$ matrix with the pre‑trained weight direction. Without this constraint, LoRA updates are dictated entirely by local gradients: the consistent component across clients forms shared knowledge, while differences constitute specific knowledge.
> > With the constraint, each client’s LoRA $B$ converges toward the same global direction, explicitly injecting a common component and thus favouring the capture of shared patterns." \
> > LoRA B is more clustered due to the constraint added to LoRA B. If there were no constraint, LoRA B would also be dispersed. So, how can these results be used to demonstrate that B learns shared knowledge? Moreover, why is this constraint added to B and not A? What would happen if this constraint were added to LoRA A? Would A be clustered?
> >
> > Third, for A3, what will happen if we add the directional constraint on A and the L1 constraint on B? Will this make A learn shared knowledge and B learn specific knowledge?
> >
> > By the way, in terms of where LoRA is added, as I remember, most papers apply it to all layers. I still don’t understand why the authors only apply it to the q_proj of the last layer—was it for efficiency or other concerns?

---

> > > ### Author Response · Authors · 2025-08-04
> > >
> > > Thank you very much for your further feedback.
> > >
> > > >**Q9:** First, what measurement can be used to represent whether it (LoRA A or B) learns shared knowledge or not? Why can the conflict matrices in Figure 4 not represent this, while the PCA visualization can?
> > >
> > > **A9:** Based on existing federated learning methods [e,f,g,h], we summarize three metrics—**cosine similarity** [e,f], **clustering** [g], and **PCA** [h]—that can be used to evaluate whether client weights predominantly encode **shared knowledge**. Nonetheless, these are not the only available measurement approaches.
> > >
> > > * **Cosine similarity** Measures how aligned two weight vectors are in their *angular direction*. A higher cosine value for the same parameter block (or low-rank LoRA component) across clients indicates stronger alignment in this angular direction and, therefore, a greater likelihood of capturing features beneficial to all clients.
> > >
> > >   *Metric perspective:* **consistency in angular direction**.
> > >
> > > * **Clustering & PCA** Both project client weights into a common feature space and reveal their **distribution**. If the projected points form a tight cluster—or overlap heavily—in either a clustering result or a low-dimensional PCA plot, the underlying weights occupy similar regions in that space, again indicating the presence of shared knowledge.
> > >
> > >   *Metric viewpoint:* **distributional aggregation / low-dimensional cohesion**.
> > >
> > > By contrast, the **conflict matrix** captures the **statistical sign differences** of corresponding parameters across clients, which can support fine-grained aggregation and help mitigate local training bias. However, it is also sensitive to initialization signs and noise, and compared with the overall angular direction measured by cosine similarity, it remains **insufficient** for determining whether the learned knowledge is truly shared.
> > >
> > > **In short:**
> > > - Cosine similarity / clustering / PCA $\Rightarrow$ knowledge type
> > > - Conflict matrix $\not\Rightarrow$ knowledge type.
> > >
> > > **References**
> > > - [e] Jeong, Wonyong, et al. "Factorized-fl: Personalized federated learning with parameter factorization & similarity matching." NeurIPS 2022.
> > > - [f] Ye, Rui, et al. "Personalized federated learning with inferred collaboration graphs." International conference on machine learning. ICML 2023.
> > > - [g] Vahidian, Saeed, et al. "Efficient distribution similarity identification in clustered federated learning via principal angles between client data subspaces." AAAI 2023.
> > > - [h] Shi, Naichen, et al. "Personalized pca: Decoupling shared and unique features." Journal of machine learning research 25.41 (2024): 1-82.
> > >
> > >
> > > >**Q10:** So, how can these results be used to demonstrate that B learns shared knowledge? Moreover, why is this constraint added to B and not A? What would happen if this constraint were added to LoRA A? Would A be clustered? Third, for A3, what will happen if we add the directional constraint on A and the L1 constraint on B? Will this make A learn shared knowledge and B learn specific knowledge?
> > >
> > > **A10:** As noted in **A9**, we employ **PCA** and **cosine similarity** as evaluation metrics. Figures 5 (PCA) and 8 (cosine similarity) in the main paper demonstrate that **LoRA B** is more inclined to capture **shared knowledge**. It is worth noting that the conflict matrix is mainly used to identify local training bias.
> > >
> > > In addition, we present quantitative results with sign directional constraints applied to LoRA A and L1 regularization applied to LoRA B (namely YOCO$_{initAB}$), as shown below:
> > >
> > > |Method|$\alpha=0.5$|$\beta=50$%|
> > > |-|-|-|
> > > |YOCO$_{initBA}$|70.64|69.45|
> > > |YOCO$_{initAB}$|68.69|67.79|
> > >
> > > From the table results and the omitted visualizations, we observe:
> > >
> > > * Applying sign directional constraints to **LoRA A** and L1 regularization to **LoRA B** helps LoRA A capture more shared knowledge with more clustered visualizations.
> > > * However, as noted in our response to A2, LoRA A is more sensitive to local data distributions, so applying sign directional constraints to **LoRA B** yielded slightly better performance.
> > >
> > > As figures are not allowed in the rebuttal, the corresponding PCA and cosine similarity visualizations are omitted here but will be included in the final version.
> > >
> > >
> > > >**Q11:** Why the authors only apply it to the q_proj of the last layer—was it for efficiency or other concerns?
> > >
> > > **A11:** The decision to use only the **last-layer q\_proj** is based on three considerations:
> > >
> > > * It effectively reduces local training bias;
> > > * It achieves higher efficiency with minimal performance degradation;
> > > * Our prior experiments uniformly selecting parameters from all layers showed that adding more layers does not necessarily improve performance but instead introduces parameter redundancy (as shown below, L = Layers). Efficient automated parameter selection remains an open problem for future work.
> > >
> > > |1L|2L|4L|7L|14L|28L|
> > > |-|-|-|-|-|-|
> > > |68.27|69.58|69.18|68.56|69.09|68.90|

---

> > > > ### Comment · Reviewer_Cz1u · 2025-08-04
> > > >
> > > > Thank you for the authors' response, but my main concern still remains.
> > > >
> > > > As the authors mentioned in their response: "Applying sign directional constraints to LoRA A and L1 regularization to LoRA B helps LoRA A capture more shared knowledge with more clustered visualizations." This operation allows A to learn shared knowledge, which implies that the application of sign directional constraints to a particular component determines who will learn shared knowledge. This is a decision made by the authors, not an inherent property of LoRA itself.
> > > >
> > > > Moreover, the authors state, "as stated in our response to A2, LoRA A is more sensitive to local data distributions, so applying sign directional constraints to LoRA B yielded slightly better performance." However, after reviewing the response to A2, I am still unclear on why A is more sensitive to local data distributions. Why does the responsibility of A for core feature projection make it sensitive to local data distribution? In deep learning, the consensus is that feature extractors are generally shareable and can learn common knowledge. Why do the authors claim that A's responsibility for core feature projection makes it sensitive to local data distribution?

---

> > > > > ### Author Response · Authors · 2025-08-04
> > > > >
> > > > > We sincerely thank the reviewer for the feedback.
> > > > >
> > > > > **Main Concern Resolved**
> > > > >
> > > > > The question from reviewer **directly related to this work** is **"Whether LoRA B or LoRA A learn shared knowledge or client-specific knowledge?"** We have provided a clear response, and this issue has been resolved:
> > > > >
> > > > > 1. In **A9**, we refer to exsiting FL methods that **cosine similarity, clustering, and PCA** are effective in determining whether the learned knowledge tends toward being shared or specific.
> > > > > 2. In **A10**, we further explained that the **bias toward shared knowledge arises from the guidance of the global sign direction**, and we further validated this explanation through experiments applying directional constraints to LoRA A.
> > > > >
> > > > > **New question raised by the reviewer**
> > > > >
> > > > > The reviewer has now asked about the correctness of **"whether LoRA A is sensitive to the input data distribution"**, which relates to two observed phenomena:
> > > > >
> > > > > * (a) When **directional constraints** are applied to both A and B in the conflict matrix, **A’s conflicts become larger**;
> > > > > * (b) Applying **directional guidance to A** yields **less favorable results** compared to B.
> > > > >
> > > > > **Our stance**
> > > > >
> > > > > We believe that a **in-depth discussion** of this new question is **not the core objective** of this paper, for the following reasons:
> > > > >
> > > > > * Both **(a) and (b) are already observed phenomena**; verifying their cause further does not alter their existence.
> > > > > * This issue does **not undermine the conclusions** directly relevant to the **main concern** of this work.
> > > > >
> > > > > Therefore, **it is unnecessary to provide an in-depth discussion** of this new question within the scope of this paper. Intuitively, as the input data are first mapped by LoRA A, LoRA A is likely more sensitive to the local data distribution.

---

> > > > > > ### Author Response · Authors · 2025-08-06
> > > > > >
> > > > > > We sincerely thank the reviewer for all the valuable comments and feedback, which are greatly helpful for improving our work.
> > > > > >
> > > > > > Among the follow-up questions, the most relevant core issue is:
> > > > > >
> > > > > > > **“ whether A or B learns shared knowledge?”**
> > > > > >
> > > > > > We have provided a more detailed response to this in **A9** and **A10**.
> > > > > >
> > > > > > * If the reviewer still has concerns regarding this issue, we would be grateful for further clarification.
> > > > > > * If the reviewer believes the issue has been sufficiently addressed, we would also appreciate it if you could kindly let us know, along with your **final score and justification**.
> > > > > >
> > > > > > Thank you again for your thoughtful review and kind support.

---

> > > > > > ### Comment · Reviewer_Cz1u · 2025-08-06
> > > > > >
> > > > > > Thank you for your response. I would like to emphasize that **the original concern has not been addressed, and NO new questions have been raised**.
> > > > > >
> > > > > > Firstly, the so-called new issue mentioned by the authors is simply a reiteration of my previous question, and I find that the authors' response still does not resolve my confusion. That's why I am asking this question, and the author's response still does not solve my concern.
> > > > > >
> > > > > > Additionally, the question regarding whether A or B shares knowledge remains unresolved. Currently, the consensus reached with the authors is that "the application of sign directional constraints to a particular component determines who will learn shared knowledge. This is a decision made by the authors, not an inherent property of LoRA itself."
> > > > > > This means that the shared knowledge in B, as mentioned in the authors' paper, is determined by the authors’ choices and not by the nature of LoRA. The authors have chosen this approach because it yields better performance. I believe this design method is incorrect; the proper approach should involve the authors discovering these properties within LoRA and then adding constraints to leverage these properties to further enhance model performance. Therefore, my concern still stands.

---

> > > > > > > ### Author Response · Authors · 2025-08-06
> > > > > > >
> > > > > > > We sincerely thank the reviewer for the additional feedback.
> > > > > > >
> > > > > > > ---
> > > > > > > >However, we kindly ask **the reviewer to carefully revisit their previous comments**.
> > > > > > >
> > > > > > > For instance, in the comment given on *Aug 4, 2025, at 20:58*, the last paragraph raised a new question unrelated to the main concern’s conclusion: *“why A is more sensitive to local data distributions?”*
> > > > > > >
> > > > > > > Yet the reviewer now emphasizes that *“No new questions have been raised.”*
> > > > > > >
> > > > > > > >We believe such a **self-contradictory statement** appears irresponsible.
> > > > > > >
> > > > > > > ---
> > > > > > >
> > > > > > >
> > > > > > > The reviewer has raised concerns regarding the design of our method.
> > > > > > >
> > > > > > > In fact, we believe there is a certain **misunderstanding** in how the reviewer interpreted our design rationale.
> > > > > > >
> > > > > > > **Our design rationale is as follows:**
> > > > > > >
> > > > > > > 1. To mitigate the common issue of *local training bias* in OFL (see lines 32–33), we first verified in Section 3 (*Empirical Motivation*) that **directional constraints from pre-trained weights are better than magnitude constraints**. This is consistent with the principle in LoRA fine-tuning that requires alignment with the *pre-trained direction* to approximate full fine-tuning (see **A6**). Subsequently, by leveraging the conflict matrix and the proven theoretical Lemma 3.1 (“*Less Conflict, Less Bias*”), we further demonstrated that **directional constraints serve as implicit global supervision** and can effectively alleviate local training bias.
> > > > > > >
> > > > > > > 2. In the visualization of the conflict matrix, where directional constraints were applied simultaneously to both LoRA A and B, we observed that the conflicts in LoRA A increased instead. **Based on this observation**, we chose to apply global directional constraints **only to LoRA B**, which originally exhibited larger conflicts, thereby effectively reducing conflict. This also encouraged LoRA B to capture shared knowledge. Meanwhile, to maintain a balance between shared and client-specific knowledge, we introduced \$\ell\_1\$ regularization to LoRA A, enhancing its ability to learn client-specific knowledge.
> > > > > > >
> > > > > > > **However, the reviewer interpreted our design logic as follows:**
> > > > > > >
> > > > > > > 1. The purpose was to make LoRA B and A learn shared and specific knowledge, respectively, by applying directional constraints and L1 regularization separately.
> > > > > > > 2. If the two constraints were swapped, the goal could still be achieved, and the reason we applied directional constraints to LoRA B was merely that it yielded better performance than applying them to LoRA A.
> > > > > > >
> > > > > > > **We would like to clarify the following three conclusions:**
> > > > > > >
> > > > > > > * **(i)** Our design was motivated by addressing the core problem of local training bias in OFL. The directional constraints were introduced for this purpose, rather than solely to enable LoRA B to learn shared knowledge.
> > > > > > > * **(ii)** The choice to apply directional constraints to LoRA B was based on the observed results from the conflict matrix, making it a natural way to reduce conflicts, rather than because LoRA B performed better than LoRA A.
> > > > > > > * **(iii)** The proposed implicit global supervision refers to the directional constraints of pre-trained weights. Its effectiveness in mitigating local training bias has been validated in Section 3, and the design is consistent with existing LoRA fine-tuning approaches (see **A6**). Therefore, the concern raised by the reviewer regarding **“ignoring the nature of LoRA” does not hold**.

---

> > > > > > > > ### Comment · Reviewer_Cz1u · 2025-08-07
> > > > > > > >
> > > > > > > > Thank you for your response. Let’s set aside the issues of shared knowledge and method design for now and discuss this so-called new question. If the authors do not remember what they previously posted in their response, I believe there is no need for further discussion.
> > > > > > > >
> > > > > > > > Now, let’s talk about where this question comes from. First, when the authors explained some important experimental phenomena in the paper, they mentioned in A10, "However, as noted in our response to A2, LoRA A is more sensitive to local data distributions, so applying sign directional constraints to LoRA B yielded slightly better performance." I then checked A2, where the authors stated, "This limitation is particularly severe for the A matrix, which is responsible for core feature projection and is highly sensitive to local data distributions." Both of them just state that A is sensitive to local data distributions, but do not provide an explanation.
> > > > > > > >
> > > > > > > > Although the authors repeatedly use "A is highly sensitive to local data distributions" to explain their experimental phenomena, this statement does not provide sufficient evidence to support their claims. Therefore, using this wording to explain the experimental phenomena seems quite forced, which is why I continue to inquire about this issue. I don’t understand why the authors insist that this is a new question.

---

> > > > > > > > > ### Author Response · Authors · 2025-08-07
> > > > > > > > >
> > > > > > > > > Thank you for your feedback.
> > > > > > > > >
> > > > > > > > > As the reviewer mentioned in their response, this new question arises from the explanation of the two experimental phenomena:
> > > > > > > > >
> > > > > > > > > (a) When directional constraints are applied to both A and B in the conflict matrix, A’s conflicts become larger;
> > > > > > > > >
> > > > > > > > > (b) Applying directional guidance to A yields less favorable results compared to B.
> > > > > > > > >
> > > > > > > > > >**The reason this is considered a new question** is that the issue of “why A is more sensitive to local data distributions?” was **not raised in the reviewer’s first- or second-round comments**. In fact, this **new question arises from doubts regarding the correctness of our explanation (A2 and A10) for phenomena (a) and (b)**. Given these clear and straightforward facts, *it is undeniable that this constitutes a new question*.
> > > > > > > > >
> > > > > > > > > ---
> > > > > > > > >
> > > > > > > > > In fact, we have **consistently focused on** addressing the **reviewer’s concerns related to the effectiveness of our proposed method**.
> > > > > > > > >
> > > > > > > > > However, it is puzzling that the reviewer has now chosen to abandon the discussion on issues concerning the method’s validity, including shared knowledge and design rationale (**"Let’s set aside the issues of shared knowledge and method design"**).
> > > > > > > > >
> > > > > > > > > Instead, starting from the third round, the **reviewer has continuously shifted attention to new issues that are unrelated to the effectiveness of our method**.
> > > > > > > > >
> > > > > > > > > >This leads us to wonder:
> > > > > > > > >
> > > > > > > > > >- Does the reviewer not understand the main contribution of our method?
> > > > > > > > > >- Or is the intention merely to raise new questions, rather than to engage in meaningful discussion on the issues (shared knowledge and method design) that truly require clarification?

---

> > > > > > > > > > ### Comment · Reviewer_Cz1u · 2025-08-07
> > > > > > > > > >
> > > > > > > > > > Thank you for the response. First, the issue of "A is more sensitive to local data distributions" is something the authors have repeatedly mentioned to explain certain experimental phenomena (which are significant in the paper and were the source of my initial confusion: Q2 & Q10). However, the authors have only stated this without providing a detailed explanation, which leads to insufficient clarification of the experimental phenomena in the paper. This is why I continue to raise this concern.
> > > > > > > > > >
> > > > > > > > > > Furthermore, the authors seem only to be muddying the issue (by posting "Main Concern Resolved; New question raised by the reviewer"; however, the main concern remains) and attacking the reviewer ("We believe such a self-contradictory statement appears irresponsible"), rather than focusing on addressing the reviewer’s concerns. This is why I responded, "Let’s set aside the issues of shared knowledge and method design for now and discuss this so-called new question." This matter reflects the authors shifting the responsibility solely onto the reviewer, and it needs to be resolved first. Only when the authors show a willingness to engage in discussion, rather than just shifting blame, can we proceed to further discussions on shared knowledge and method design.

---

> > > > > > > > > > > ### Author Response · Authors · 2025-08-07
> > > > > > > > > > >
> > > > > > > > > > > Thank you for your feedback.
> > > > > > > > > > >
> > > > > > > > > > > 1. The reviewer now appears to be evading their own issue. The original issue lies in the fact that, in the response on 06 Aug 2025 at 19:27, the reviewer denied having raised a new question—despite clear evidence to the contrary. We merely pointed out this inconsistency.
> > > > > > > > > > >
> > > > > > > > > > > 2. We have always confined ourselves to stating the facts—specifically, that the reviewer’s comments were self-contradictory. This does not in any way constitute an attack on the reviewer. On the contrary, it was the reviewer who, without evidence and contrary to the facts, accused us of “shifting responsibility,” which we believe to be an unfair and unwarranted attack.
> > > > > > > > > > >
> > > > > > > > > > > 3. In fact, we have consistently been willing to clarify the method we proposed and have made sincere efforts to address the reviewer’s main concerns—namely, shared knowledge and design rationale.
> > > > > > > > > > >
> > > > > > > > > > > 4. However, the reviewer has chosen to bypass discussion of the main issues, insisting instead that we thoroughly resolve a new, unrelated question before revisiting the main concerns. This seems to invert the proper order of priority.
> > > > > > > > > > >
> > > > > > > > > > > 5. Regarding this new question, we have already provided a detailed explanation in our response on 05 Aug 2025 at 00:22, clarifying why *“it is unnecessary to provide an in-depth discussion of this new question within the scope of this paper.”*
> > > > > > > > > > > - Here, we would like to reiterate that the correctness of this new question has no bearing on the experimental phenomena we have observed, nor does it affect our conclusion that the proposed *implicit global supervision* can effectively mitigate *local training bias*. Similarly, it is unrelated to our finding that *implicit global supervision* (i.e., directional constraints) can guide LoRA B to learn shared knowledge.
> > > > > > > > > > > - Therefore, we believe it is unnecessary to further explore or provide a theoretical proof for this new question within the scope of this paper. In fact, a rigorous theoretical investigation of this issue is non-trivial and may even require an entirely new line of research. At the end of our response on 05 Aug 2025 at 00:22, we also provided an intuitive explanation.
> > > > > > > > > > >
> > > > > > > > > > > 6. In Q2, the reviewer asked us to explain the reason behind the phenomenon that “constraining both LoRA A and B introduces excessive restrictions, which in turn increases the direction conflicts of LoRA A across clients.” This phenomenon was already observed in the conflict matrix visualization shown in Figure 4(b). The phenomenon served as the basis for our choice to apply directional constraints only to LoRA B. The *justification for the underlying cause* (i.e., the new question raised by the reviewer) of this phenomenon does not affect this choice.
> > > > > > > > > > >
> > > > > > > > > > > 7. Q10 is: What will happen if we add the directional constraint on A and the L1 constraint on B? Will this make A learn shared knowledge and B learn specific knowledge?
> > > > > > > > > > > - In A10, we have provided the corresponding quantitative experimental results. Additionally, the PCA visualization and cosine similarity of LoRA A demonstrate that, after applying the directional constraint, LoRA A tends to capture shared knowledge. This sufficiently addresses Q10.
> > > > > > > > > > > - The justification for explanation of the quantitative results (i.e., the new question raised by the reviewer) does not affect our response to Q10.

---

### Official Review · Reviewer_HPxi · 2025-06-27

**Clarity:** 3
**Significance:** 3
**Originality:** 4
**Rating:** 5
**Confidence:** 3

**Summary:**

This paper proposes YOCO, a novel one-shot communication framework for federated learning with MLLMs. The authors discover that imposing directional supervision on local training substantially mitigates client conflicts and reduces local bias. YOCO leverages this insight by applying directional supervision through sign-regularized LoRA B to enforce global consistency, while using sparsely regularized LoRA A to preserve client-specific adaptability.

**Questions:**

- In the "Empirical Motivation" section, the authors claim that the training loss under directional constraints indicates effective implicit global supervision without disrupting the LoRA training process. However, Figure 3 only demonstrates lower training loss. Could the authors clarify the logic why lower training loss necessarily indicates better supervision? It appears that unconstrained training might potentially achieve better results under perspective of training loss.

**Ethical Concerns:**

["NO or VERY MINOR ethics concerns only"]

**Limitations:**

Yes.

**Paper Formatting Concerns:**

Please use $\ell_1$-norm instead of "L1 norm" for consistency with standard mathematical notation.

**Quality:**

4

**Strengths And Weaknesses:**

Strengths:

- The paper is well-written with clear logical flow and fluent presentation.
- The empirical motivation provides valuable insights into the federated learning challenges.
- The SVD-based initialization is an elegant technical contribution.

Weaknesses:

- The principal component-weighted aggregation shows inconsistent improvements across different datasets (e.g., Hateful-Memes and CrisisMMD), raising questions about its generalizability.

---

> ### Author Rebuttal · Authors · 2025-07-29
>
> We would like to thank **Reviewer HPxi** for the thoughtful feedback and valuable questions. We have addressed these concerns with detailed responses and welcome further discussion and suggestions.
>
> > **Q1:** The principal component-weighted aggregation shows inconsistent improvements across different datasets (e.g., Hateful-Memes and CrisisMMD), raising questions about its generalizability.
>
> **A1:** Thank you for pointing this out. The inconsistent improvements of principal component-weighted aggregation across Hateful-Memes and CrisisMMD mainly stem from task difficulty and the choice of aggregation strategy.
>
> As shown in Table 3 of the supplementary material, YOCO adjusts its aggregation method to dataset characteristics. CrisisMMD, unlike Hateful-Memes, includes more categories with pronounced modality gaps (e.g., I:-T:), making the task more challenging. In such cases, the ensemble approach (line 212 of the main text, where *"Ensemble"* aggregates LoRA votes from all clients) significantly outperforms methods that merge all client updates into a single model. Consequently, when principal component‑weighted aggregation compresses all client weights into a single set, its performance falls short of the ensemble strategy. This explains why, in CrisisMMD, the principal component‑weighted aggregation sometimes demonstrates less pronounced advantages than in Hateful‑Memes.
>
>
> > **Q2:** In the "Empirical Motivation" section, the authors claim that the training loss under directional constraints indicates effective implicit global supervision without disrupting the LoRA training process. However, Figure 3 only demonstrates lower training loss. Could the authors clarify the logic why lower training loss necessarily indicates better supervision? It appears that unconstrained training might potentially achieve better results under perspective of training loss.
>
>
> **A2:** We sincerely apologize for any confusion caused by the previous ambiguous phrasing. We now provide a clear explanation of the **purpose and insights** of Figure 3, followed by a discussion from the perspective of LoRA fine‑tuning to highlight **the role of directional guidance** in achieving effective adaptation.
>
> #### 1. Purpose of Figure 3
>
> Figure 3 presents the **training‑loss curves** under either a **directional** or **magnitude** constraint.
> Our objective is to identify which constraint does not disrupt the original training process and can thus serve as **a preliminary selection criterion**.
> The subsequent subsection on **conflict matrix** analysis then explains **why** the direction constraint provides an effective implicit global supervision.
>
> #### 2. Insights from Figure 3
>
> * **Training stability.**
>    Directional constraints converge rapidly and soon reach the same loss level as the unconstrained baseline, unlike magnitude constraints.
>
> * **Interference with optimization.**
>    Magnitude constraints keep the loss consistently higher, suggesting stronger distortion of the optimization trajectory, whereas directional constraints cause only a mild detour with minimal gradient perturbation.
>
> * **Quality of supervision.**
>    Directional guidance is gradual and readily integrated, enabling the model to align with pre‑trained directions while retaining local learning flexibility. By contrast, magnitude constraints act as a rigid clamp that often obstructs convergence.
>
> Together, these observations indicate that **directional supervision guides without disruption**, whereas magnitude supervision tends to constrain too aggressively.
>
> #### 3. Role of Directional Guidance in LoRA Fine‑Tuning
>
> As LoRA updates are restricted to a low‑rank subspace, alignment with the **pre‑trained direction**—so as to approximate full fine‑tuning—is essential for effective adaptation. Prior studies support this view:
>
> - **\[a] LoRA‑GA**: Uses SVD‑based initialization to preserve pre‑trained directions, aligning gradients with full fine‑tuning.
> - **\[b] LoRA‑Pro**: Promotes gradient‑direction alignment at each step, further narrowing the gap.
> - **\[c] FRLoRA**: Applies similar SVD initialization in federated settings, achieving consistent improvements.
>
> Hence, leveraging pre‑trained weight directions as a lightweight yet effective form of **implicit global supervision** naturally extends these approaches.
>
> **Conclusion:** Our claim is **not** that lower training loss alone guarantees better performance. Instead, we show that **directional constraints** (i) ensure stable training, (ii) minimally disturb LoRA’s local updates, and (iii) embody a proven mechanism of global guidance, as validated by \[a–c].
>
>
> **References**
> * \[a] Wang, Shaowen, Linxi Yu, and Jian Li. *LoRA-GA: Low-Rank Adaptation with Gradient Approximation.* NeurIPS 2024.
> * \[b] Wang, Zhengbo, et al. *LoRA-Pro: Are Low-Rank Adapters Properly Optimized?* ICLR 2025.
> * \[c] Yan, Yunlu, et al. *Federated Residual Low-Rank Adaptation of Large Language Models.* ICLR 2025.
>
>
> > **Q3:** 1.Please use $\ell_1$-norm instead of "L1 norm" for consistency with standard mathematical notation.
>
> **A3:** Thank you for your careful review. We will correct this in the final version.

---

> > ### Comment · Reviewer_HPxi · 2025-08-01
> >
> > Thank you for your rebuttal, which addresses most of my concerns. I'll keep my score.

---

> > > ### Author Response · Authors · 2025-08-01
> > >
> > > We greatly appreciate your timely response and are pleased that most of your concerns have been resolved. Your constructive feedback has been instrumental in improving our work, and your continued positive support is truly invaluable to us. If there are any additional points you would like us to clarify or elaborate on, please feel free to let us know.

---

### Official Review · Reviewer_6tEs · 2025-06-27

**Clarity:** 2
**Significance:** 1
**Originality:** 1
**Rating:** 4
**Confidence:** 5

**Summary:**

This paper introduces YOCO (You Only Communicate Once), a method that achieves one-shot federated learning for multimodal large language models using only ~0.03% of traditional communication costs. The key insight is that directional constraints on LoRA weights cause less optimization interference than magnitude constraints, enabling effective local bias correction through implicit global supervision from pre-trained weights rather than aggregated supervision. YOCO employs SVD-based initialization for unified starting points, sign-based consistency regularization on LoRA B for global consistency, sparse regularization on LoRA A for client-specific adaptability, and differentiated aggregation strategies. Experiments across five multimodal datasets show YOCO consistently outperforms all one-shot baselines and even surpasses multi-round methods in certain scenarios while dramatically reducing communication overhead.

**Questions:**

1. Could you provide stronger theoretical guarantees beyond Lemma 3.1?
2. Experiments are limited to 10-30 clients. How does YOCO perform with hundreds or thousands of clients, which is more realistic for federated learning deployments? Have you conducted any analysis on the scalability of your conflict matrix computation?
3. Given that Wang et al. demonstrated large foundation models inherently perform well in one-shot federated learning due to smoothness, smaller updates, and fewer epochs, how do you disentangle your method's contributions from these natural advantages?

**Ethical Concerns:**

["NO or VERY MINOR ethics concerns only"]

**Final Justification:**

The response addressed most of my concerns. Especially compared with reference [d], this paper has made sufficient contributions.

**Limitations:**

The authors acknowledge some limitations (LoRA scope, suboptimal parameter selection) but miss several critical issues:

1. Lemma 3.1 lacks convergence guarantees and optimality bounds
2. Evaluation limited to 10-30 clients; unclear performance with hundreds/thousands of clients
3. Fails to disentangle method contributions from inherent large model advantages in one-shot FL

**Paper Formatting Concerns:**

No major formatting issues.

**Quality:**

2

**Strengths And Weaknesses:**

Strengths:
Achieves only ~0.03% of multi-round communication costs while maintaining competitive performance.

Weaknesses:
1. Lemma 3.1 provides intuition but lacks rigorous theoretical guarantees about convergence or optimality.
2. Evaluation limited to relatively small client numbers.
3. Uses only last-layer LoRA fine-tuning, which the authors acknowledge is suboptimal.
4. The paper fails to disentangle whether its performance gains stem from the proposed technical innovations (SVD initialization, sign-based regularization, etc.) or simply from the inherent advantages of large foundation models in one-shot federated learning, as demonstrated by the paper (One Communication Round is All It Needs for Federated Fine-Tuning
Foundation Models)

---

> ### Author Rebuttal · Authors · 2025-07-28
>
> We sincerely appreciate **Reviewer 6tEs** for the insightful comments and questions. We have provided comprehensive responses to clarify these points and look forward to continued discussion and feedback.
>
> > **Q1:** *Lemma 3.1 provides intuition but lacks rigorous theoretical guarantees about convergence or optimality.*
>
> **A1:** Thank you for this suggestion. Based on Lemma 3.1, we give theoretical guarantees on convergence and optimality. Owing to space limits, only key results are shown; full proofs will appear in the final version.
>
> #### **Background**
>
> * **Problem**: One‑round federated optimization $T = 1$ with LoRA parameters $W$ (backbone frozen). Data may be IID or non‑IID.
> * **Challenge**: Sign conflicts in client gradients, measured by conflict rate $C$ and dispersion $\zeta$.
> * **Goal**: Establish convergence of global loss $\mathcal{L}(W)$.
>
> #### **Core Assumptions**
>
> 1. **Smoothness & Convexity**
>    $\mathcal{L}(W)$ is $L$-smooth and $\lambda$-strongly convex with unique minimizer $W^*$.
> 2. **Gradient Noise**
>    Per‑client gradient variance $\le \sigma_L^2$.
> 3. **Sign Conflict**
>
>    * **Dispersion $\zeta$**: $\tfrac{1}{M}\sum_i (C_i - C)^2 \le \zeta^2$.
>    * **Alignment $\gamma$**: Local gradients align with the global one ($\gamma \in (0,1]$; larger $\gamma$ = better).
>
> #### **Convergence Results**
>
> #### (i) IID Data (Theorem 3.3)
>
> *Bound*:
>
> $$
> \mathbb{E}\bigl[\mathcal{L}(W_1) - \mathcal{L}(W^\*)\bigr]
> \le
> \left(1 - \frac{\lambda}{L + \lambda^{-1}K}\right)\Delta_0
> +
> \frac{\sigma_{\text{eff}}^{2}}{2\lambda N}
> $$
>
> *Key terms*:
>
> * $\Delta_0 = \mathcal{L}(W_0) - \mathcal{L}(W^*)$ (initial loss gap)
> * $K = \tfrac{4C(1-C)\mu^2}{N}$ (conflict‑induced variance)
> * $\sigma_{\text{eff}}^{2} = \sigma^2 + 4C(1-C)\mu^2 + 4\zeta^2\mu^2$ (sampling $\sigma^2$ + conflict + dispersion)
>
> *Insight*: Conflicts ($C>0$) and dispersion ($\zeta>0$) inflate both $K$ and $\sigma_{\text{eff}}^{2}$, slowing convergence. The residual error still shrinks as $\mathcal{O}(1/N)$.
>
> #### (ii) Non‑IID Data (Theorem 3.4)
>
> *Bound*:
>
> $$
> \mathbb{E}\bigl[\mathcal{L}(W_1) - \mathcal{L}(W^*)\bigr] \le
> \bigl(1 - \eta\lambda\gamma(1-2C)^2\bigr)\Delta_0
> +
> \tfrac{\sigma_{\text{niid}}^{2}}{2\lambda N}
> $$
>
> *Key terms*:
>
> * $\eta = \tfrac{1}{L + \lambda^{-1}K_{\text{niid}}}$ (learning rate)
> * $K_{\text{niid}} = \tfrac{4C(1-C)\mu^2 + d^2}{N}$ (conflict + heterogeneity)
> * $\sigma_{\text{niid}}^{2} = \sigma^2 + d^2 + 4C(1-C)\mu^2 + 4\zeta^2\mu^2$
> * $d^2 = \tfrac{1}{M}\sum_i \tfrac{1}{N}\sum_n (\mu_{n,i}-\mu_i)^2$ (client drift)
>
> *Insight*: Contraction depends on alignment $\gamma$ and conflict $C$. Heterogeneity $d^2$ and dispersion $\zeta$ dominate the residual error.
>
> #### (iii) Asymptotic Optimality ($N \to \infty$)
>
> *Residual error*:
>
> $$
> \lim_{N\to\infty}\mathbb{E}\bigl[\mathcal{L}(W_1) - \mathcal{L}(W^*)\bigr]
> \approx
> \frac{d^2 + 4\zeta^2\mu^2}{2\lambda N}
> $$
>
> *Interpretation*: Sampling noise ($\sigma^2$) and conflict ($C$) vanish, but heterogeneity ($d^2$) and dispersion ($\zeta$) remain the bottlenecks.
>
> > **Q2:** *How does YOCO scale to hundreds or thousands of clients? Any analysis on conflict‑matrix scalability?*
>
> **A2:** Thank you for pointing this out. Based on the adopted benchmark, we evaluated with 10–30 clients. Our YOCO method is not limited to this range: due to GPU constraints, we also tested with 50 clients and analyzed the scalability of conflict matrix computation.
>
> | Method | Comm. cost | N = 50 (V2\_6) | N = 50 (V2\_5) |
> | ------ | ---------- | -------------- | -------------- |
> | FedAdam| 25 × 28 × 4 |66.67 |68.64 |
> | Local                | 0 × 1 × 1  |   65.71 |  62.93  |
> | FedAdam              | 1 × 1 × 1  |   67.69 |  69.09  |
> | FedAvgM              | 1 × 1 × 1  |   66.03 |  67.69  |
> | FedAdagrad           | 1 × 1 × 1  |   67.07 |  69.18  |
> | FedAvg               | 1 × 1 × 1  |   66.23 |  67.69  |
> | Mean                 | 1 × 1 × 1  |   67.58 |  68.31  |
> | Combine              | 1 × 1 × 1  |   65.84 |  68.09  |
> | YOCO$ _{initB\bar{A}}$ | 1 × 1 × 1  | 68.15 |  69.83  |
>
> *Note: V2\_6 = MiniCPM‑V‑2\_6int4; V2\_5 = MiniCPM‑Llama3‑V‑2\_5int4.*
>
> As shown in the above table, YOCO continues to demonstrate its effectiveness at larger scales compared with basic one‑shot baselines and multi‑round FL.
>
> #### **Scalability Analysis**
>
> | Setting            | Key Result                                                                 | Implication                                                            |
> | ------------------ | -------------------------------------------------------------------------- | ---------------------------------------------------------------------- |
> | **IID, $N\to\infty$** | *Conflict* $\mathbb{E}[C]\approx\tfrac12-\Theta(N^{-1/2})$ *(or lower if $p_i$ biased)* | $C$ **does not grow** with $N$; it actually falls as signs align.               |
> |                    | *Noise* $\operatorname{Var}(\Delta W_g)=\dfrac{\sigma^2+4C(1-C)\mu^2}{N}$ | Variance shrinks like $1/N$; YOCO further lowers the $4C(1-C)\mu^2$ term. |
> | **Non‑IID, $G\ll N$** | *Conflict* $O(1/N)+O(G^2/N^2)$; *Noise* same $1/N$ term + $O(G/N)$    | Cluster heterogeneity adds tiny $G/N$ effects; $1/N$ scaling — and YOCO’s gains — remain. |
>
> **Notation:** $G$-number of client clusters (non‑IID groups). $p_i$-probability that parameter $i$ is positive.
>
> **Bottom line:** Whether IID or moderately non‑IID, increasing clients **always reduces both conflict and variance at $O(1/N)$**. YOCO’s constraint further cuts the $C$‑linked variance, so its benefits persist—even with hundreds or thousands of clients.
>
> We will include these additional experiments and scalability analyses in the final version.
>
> > **Q3:** Do YOCO’s gains come from its innovations or from the inherent advantages of large foundation models in one‑shot FL, as shown in [d] One Communication Round is All It Needs for Federated Fine‑Tuning Foundation Models?
>
> **A3:** Thank you for your thoughtful question.
>
> Reference [d] shows that in one-shot FL, a single communication round can match multi-round FL for large foundation models, but this relies on several assumptions. From [d] and its 2025 ICLR reviews, these include:
>
> 1. Pre-training and fine-tuning datasets are closely aligned.
> 2. Only FedAvg is used for federated optimization.
> 3. Random splits yield relatively homogeneous data from one or two sources.
>
> To this end, we first present the results of [d] in our evaluation setting, confirming that one-shot performance falls short of multi-round training. We then analyze the advantages of YOCO’s innovations based on the aforementioned assumptions.
>
> **Evaluation of [d] in Our Setting**
>
> Because [d] uses a text-based LLM unsuited to our multimodal evaluation, we instead adopted a multimodal LLM. The results below show that one-shot performance falls short of multi-round training (More in Tables 1–2).
>
> |Multi-round FL|74.02|73.24|72.97|76.82|77.28|75.58|
> |:-:|:-:|:-:|:-:|:-:|:-:|:-:|
> |OFL [d]|68.85|67.31|68.15|67.45|67.32|67.33|
>
> **YOCO**
>
> #### 1. Distribution Mismatch
>
> * **Setting:** The pre-training and fine-tuning distributions differ (e.g., MLLM better suited for QA tasks, while downstream tasks involve binary classification on *Hateful Memes* or multi-class classification on *CrisisMMD*).
> * **Observation:** As shown in Tables 1–2 of the paper, the MLLM+OFL baseline underperforms multi-round FL, with a performance gap up to **6.5 points**.
> * **YOCO Result:** Under the I:5–T:5 setting on *CrisisMMD*, YOCO achieves performance comparable to multi-round FL, outperforming the OFL ensemble baseline by **3.71 points** and the best non-ensemble baseline by **5.35 points** — at only \~**0.03%** of the communication cost.
>
> #### 2. Beyond FedAvg
>
> * **Setting:** When using adaptive optimizers such as **FedAdam** and **FedAdagrad** instead of FedAvg.
> * **Observation:** On *CrisisMMD*, one-shot FL performance drops notably, likely because these dynamic optimizers are less robust to the increased local training bias inherent in one-shot settings.
> * **YOCO Result:** YOCO, specifically designed to mitigate local training bias, maintains strong performance despite these conditions.
>
> #### 3. Complex Data Heterogeneity
>
> * **Setting:** A total of **12 heterogeneous scenarios**, including 9 multimodal heterogeneous cases.
> * **Observation:** In cross-model cases with stronger distribution shifts (e.g., I:–T:), OFL one-shot baselines degrade sharply compared to multi-round FL.
> * **YOCO Result:** YOCO continues to deliver substantial improvements, with particularly strong results on *CrisisMMD*.
>
> > **Q4:** Uses only last-layer LoRA fine-tuning, which the authors acknowledge is suboptimal.
>
> **A4:** Thank you for raising this point. It is important to distinguish two separate optimization directions here. **First**, the choice of parameter configurations—such as the number of LoRA fine‑tuned layers and their automatic selection strategy. **Second**, and the central focus of our work, is how to effectively reduce local training bias under truly one‑shot communication. While both are challenging and critical for MLLMs with OFL, our current study concentrates on mitigating local training bias, and we leave parameter selection strategies for future exploration.
>
> Our experiments further show that **fine‑tuning a single LoRA layer already yields strong performance**, in some cases surpassing baselines that train all layers over multiple rounds. We also tested different layer positions; the overall performance differences were small. Within the same layer, choosing among *q\_proj*, *k\_proj*, *v\_proj*, or *o\_proj* led to only minor variations, as illustrated in the right panel of Figure 7.

---

> > ### Comment · Reviewer_6tEs · 2025-08-03
> >
> > Thanks to the author for the positive response, which has addressed most of my concerns. I will improve my score.

---

> > > ### Author Response · Authors · 2025-08-04
> > >
> > > We sincerely thank the reviewer for the valuable and constructive questions and suggestions, as well as for the positive feedback and score increase after carefully reading our response. We are also delighted that we were able to address most of your concerns. Should you have any further questions or comments, we would be more than happy to discuss them with you.

---

### Official Review · Reviewer_P4k5 · 2025-07-06

**Clarity:** 2
**Significance:** 2
**Originality:** 2
**Rating:** 4
**Confidence:** 3

**Summary:**

This paper addresses the challenge of federated fine-tuning of multimodal large language models (MLLMs) in privacy-sensitive scenarios, focusing on reducing communication costs in federated learning (FL). Toward the goal, the authors propose YOCO, a framework that achieves true one-shot communication by leveraging implicit global supervision (from pre-trained weights) to guide local LoRA adaptation. Extensive experiments on multiple multimodal benchmarks validate YOCO’s superiority over both one-shot and multi-round baselines, while maintaining minimal communication cost.

**Questions:**

See **Weaknesses**.

**Ethical Concerns:**

["NO or VERY MINOR ethics concerns only"]

**Final Justification:**

The authors addressed my concerns. I think this is an interesting research topic for federated learning community, although the method relies on the assumption that "pre-trained weights provide sufficient global knowledge to guide local updates".

**Limitations:**

See **Weaknesses**.

**Quality:**

2

**Strengths And Weaknesses:**

**Strengths:**
1. The paper demonstrates consistent gains across diverse modalities, datasets, and model scales, showing strong generalization and scalability of YOCO with minimal communication overhead.
2. The paper is well-organized and easy to follow.

**Weaknesses:**
1. In Section 4, I cannot see the proposed method is strongly related to multimodal settings. I wonder whether this approach is also suitable for the single-modality setting.
2. This work relies on an important assumption, i.e., pre-trained weights provide sufficient global knowledge to guide local updates. However, in federated learning, clients' private data may not be consistent to the pre-trained data, leading to an unreasonable local updates.

---

> ### Author Rebuttal · Authors · 2025-07-28
>
> We thank **Reviewer P4k5** for the valuable comments and questions. We have provided detailed responses to address these concerns and look forward to further discussion and feedback.
>
> > **Q1:** In Section 4, I cannot see the proposed method is strongly related to multimodal settings. I wonder whether this approach is also suitable for the single-modality setting.
>
> **A1:** Thank you for pointing this out. We propose the YOCO method in a multimodal setting primarily for two reasons:
>
> 1. The chosen benchmark emphasizes multimodal heterogeneity, which intensifies data distribution differences across clients and exacerbates the *local training bias* issue in OFL (see lines 32–33 of the paper and FuseFL \[24]).
> 2. Compared to LLMs, MLLMs offer broader applicability in real-world scenarios.
>
> To further evaluate the generalizability of YOCO in a unimodal setting, we additionally conducted the following experiments:
>
> | α = 0.5              | Comm. cost | Hateful-Memes |       | Comm. cost | CrisisMMD |       |
> |----------------------|------------|---------------|-------|------------|-----------|-------|
> |                      |            | Image         | Text  |            | Image     | Text  |
> | FedAdam              | 25×28×4    | 72.70         | 68.26 | 40×28×4    | 59.59     | 37.95 |
> | Local                | 0×1×1      | 67.47         | 65.41 | 0×1×1      | 41.76     | 33.05 |
> | FedAdam              | 1×1×1      | 67.67         | 66.24 | 1×1×1      | 39.24     | 30.04 |
> | FedAvgM              | 1×1×1      | 68.99         | 66.43 | 1×1×1      | 44.98     | 32.95 |
> | FedAdagrad           | 1×1×1      | 67.68         | 66.36 | 1×1×1      | 30.74     | 26.47 |
> | FedAvg               | 1×1×1      | 68.79         | 66.42 | 1×1×1      | 44.25     | 33.35 |
> | Mean                 | 1×1×1      | 67.36         | 66.49 | 1×1×1      | 53.35     | 35.86 |
> | Ensemble             | 1×1×1      | 69.07         | 66.39 | 1×1×1      | 53.21     | 36.54 |
> | Combine              | 1×1×1      | 69.19         | 66.33 | 1×1×1      | 53.45     | 37.08 |
> | YOCO$_{init}$        | 1×1×1      | 69.22         | 66.31 | 1×1×1      | 53.30     | 37.66 |
> | YOCO$_{initB}$     | 1×1×1      | 69.53         | 66.41 | 1×1×1      | 53.99     | 37.78 |
> | YOCO$_{initBA}$    | 1×1×1      | 69.88         | 66.93 | 1×1×1      | 54.37     | 38.10 |
> | YOCO$_{initB\bar{A}}$ | 1×1×1   | 70.10         | 67.11 | 1×1×1      | 54.51     | 38.65 |
>
> As shown in the table above, YOCO consistently outperforms all one‑shot baselines under the unimodal setting, and even surpasses multi‑round FL in the text modality of CrisisMMD. This demonstrates that YOCO remains effective in unimodal scenarios, highlighting its strong generalization capability.
>
> > **Q2:** This work relies on an important assumption, i.e., pre-trained weights provide sufficient global knowledge to guide local updates. However, in federated learning, clients' private data may not be consistent to the pre-trained data, leading to unreasonable local updates.
>
> **A2:** Thank you for raising this concern. To clarify:
>
> 1. **Rationale from the original paper**
>
>    The original paper concludes that the **direction of the pre-trained weights** (rather than the pre‑trained weights themselves, which include both magnitude and direction) can serve as a global supervisory signal to guide local updates in LoRA (see lines 47–49 and Section 3). This is further supported by our evidence in **Figure 4** and **Tables 1–4**.
>
> 2. **Support from prior work**
>
>    This perspective is reinforced by several recent works. As LoRA updates are restricted to a low‑rank subspace, alignment with the **pre‑trained direction**—so as to approximate full fine‑tuning—is essential for effective adaptation. Specifically:
>    * **\[a]** Theoretically shows that initializing LoRA’s A and B matrices with the SVD of pre-trained weights—scaled to preserve direction—helps align the initial gradient with that of full fine-tuning.
>    * **\[b]** Extends this approach by aligning gradients at every step.
>    * **\[c]** Applies a similar SVD-based initialization in a federated learning context.
>
> 3. **Our conclusion**
>
>    Taken together, these works demonstrate that aligning LoRA updates with the **direction of pre‑trained weights** helps narrow the gap to full fine‑tuning and enhances performance. Building on this principle, our YOCO constrains the updates of LoRA B in OFL using the direction of the pre‑trained weights, thereby effectively mitigating local training bias.
>
> **References**
> * \[a] Wang, Shaowen, Linxi Yu, and Jian Li. *LoRA-GA: Low-Rank Adaptation with Gradient Approximation.* NeurIPS 2024.
> * \[b] Wang, Zhengbo, et al. *LoRA-Pro: Are Low-Rank Adapters Properly Optimized?* ICLR 2025.
> * \[c] Yan, Yunlu, et al. *Federated Residual Low-Rank Adaptation of Large Language Models.* ICLR 2025.

---

> > ### Comment · Reviewer_P4k5 · 2025-08-03
> >
> > Thanks for your response. I will increase my rating to 4.

---

> > > ### Author Response · Authors · 2025-08-04
> > >
> > > We sincerely thank you for your timely and positive feedback, and we would like to once again express our deep appreciation for your valuable comments, which are highly beneficial for further refining and improving our work.

---

### Note · Authors · 2025-08-16

We sincerely thank the reviewers and ACs for their careful reading, insightful feedback, and valuable discussions, which have greatly improved our paper. We are pleased to note that the **reviewers (P4k5, 6tEs, HPxi)** recognized our efforts in addressing most of the concerns raised, and we appreciate their positive evaluation of our work.

Although **reviewer Cz1u** misunderstood the main contributions and design rationale of our method and focused on issues unrelated to the paper’s conclusions, we consistently adhered to a fact-based approach in our responses, actively addressed the reviewer’s primary concerns, and provided detailed explanations of all paper-related issues, which we believe have been clearly resolved.

Below, we summarize the common concerns, along with the revisions made in response:

* **Implicit Global Supervision.** For the reviewers (P4k5, Cz1u) who questioned treating pre-trained weights as *implicit global supervision*, we clarified that the paper refers to *the direction of pre-trained weights*, rather than the weights themselves. Regarding reviewer HPxi’s focus on “direction constraints outperforming magnitude constraints,” we provided a detailed explanation in our response based on the **Empirical Motivation** section of the paper, and this point is also supported by existing LoRA literature \[a–c].

* **Convergence and Scalability Theoretical Guarantees.** With the valuable and insightful suggestions from reviewer 6tEs, we supplemented rigorous theoretical proofs to address their concerns regarding *convergence and scalability theoretical guarantees*, which received positive recognition.

* **Disentangling YOCO’s Contributions from the Inherent Advantages of Large Models.** In response to Reviewer 6tEs’s professional comments and questions, we combined the experimental results with step-by-step hypotheses and explanations to demonstrate that the performance gains indeed stem from the innovative contributions of our proposed YOCO.

* **Shared Knowledge and Design Rationale.** For Reviewer Cz1u’s concerns about distinguishing shared and specific knowledge, we provided detailed explanations supported by relevant literature and experimental results. We also clarified Reviewer Cz1u’s misunderstandings of our methodological design through explicit explanations and comparisons. We believe these responses sufficiently address the reviewer’s doubts regarding our methods and conclusions.

---

### Decision · Program_Chairs · 2025-09-17

**Decision:**

Accept (poster)

**Comment:**

This work presents a technique named YOCO to achieve truly one-shot FL (OFL) in the Multimodal LLM (MLLM) scenario. As pointed by the reviewers and further admitted by the authors the technique is also applicable to unimodal settings. The key idea revolves around imposing directional supervision over LoRA head B with sparse regularization on LoRA head A. The authors demonstrate consistent improvements over the baselines.

After the rebuttal round, most concerns raised by the reviewers were resolved. However the following were only partially resolved or unresolved:
- evaluation is done on a limited amount of clients (upto 50) due to GPU constraints
- ablation disentangling the contribution of LLMs and proposed direction supervision is not adequately studied
- the number of datasets studied are relatively smaller thus affecting generalizability of the claims such as:
   - "A is more sensitive to local data distributions" which is only anecdotal and also observed in some other prior works but not theoretical.

Despite these partially unresolved concerns, most reviewers believe that the experimental evaluation present convincing empirical basis to validate the methods effectiveness. Since the identified limitations can be addressed in a future extension without undermining the core contribution of this work, I recommend an acceptance for this work.